# The social costs of aviation $CO_2$ and contrail cirrus

Daniel J. A. Johansson [1] ✉, Christian Azar[1], Susanne Pettersson[1], Thomas Sterner [2], Marc E. J. Stettler [3] & Roger Teoh[3]

The radiative forcing (RF) of contrail cirrus is substantial, though short-lived, uncertain, and heterogeneous, whereas the RF from $CO_2$ emissions is long-term and more predictable. To balance these impacts, we calculate the social costs of $CO_2$ and contrail cirrus using a modified Dynamic Integrated Climate Economy (DICE) model, spanning three discount rates, two damage functions, and three climate pathways. The main case estimate of the global social cost ratio of contrail cirrus to aviation $CO_2$ emissions ranges from 0.075 to 0.57, depending on assumptions. Accounting for uncertainty in contrail cirrus RF and climate efficacy further widens this range. We also quantify flight-specific social costs of contrail cirrus by analyzing nearly 500,000 flights over the North Atlantic, revealing substantial variability due to meteorological conditions. While uncertainty is considerable, our findings suggest that carefully implemented operational contrail avoidance could offer climate benefits even when the social cost of additional $CO_2$ emissions is considered.

It has long been recognized that aviation has important climate impacts in addition to its fossil $CO_2$ emissions[1]. Currently, aviation contributes about 2.5% of the global annual $CO_2$ emissions, while its total contribution to current warming is larger due to the impact of non-$CO_2$ forcers. Most important of the non-$CO_2$ forcers is contrail cirrus, which currently has an effective radiative forcing (ERF) comparable to that of aviation $CO_2$[2].

Contrail cirrus is formed when hot and moist aircraft exhaust meets ambient air that is cold and humid enough[3]. In the formation process exhaust water vapor condenses on particles originating primarily from soot in the aircraft exhaust[4]. For the contrails to remain in the atmosphere for more than a few minutes, the air must be ice supersaturated. Under such conditions, atmospheric vapor continues to deposit on the ice nuclei. Eventually, the cloud sublimates. Persistent contrail cirrus has atmospheric lifetimes on the order of hours[5,6].

Contrail cirrus has both warming and cooling effects since it absorbs infrared radiation and scatter solar radiation. Consequently, contrails have a much smaller warming impact in the daytime, and they may even have a net cooling effect[7]. Further, ice-supersaturated regions tend to be variable in both space and time and hard to

forecast[8]. Hence, the properties of contrail cirrus depend on ambient conditions, fuel properties, and engine characteristics, causing substantial spatiotemporal variability and uncertainty in their formation and warming effects. At the global level, it has been estimated that about 2–3% of all flights lead to 80% of the ERF caused by all contrail cirrus[6].

$CO_2$ emissions have long-lasting effects in the atmosphere. Given prevailing climate conditions about 40% of an emission pulse is present in the atmosphere after 100 years, and about 20% after 1000 years[9]. Furthermore, the long atmospheric lifetime of $CO_2$, combined with the thermal inertia of the oceans, results in a close to linear relationship between cumulative $CO_2$ emissions and global mean surface temperature change[10]. This can be contrasted to the forcing of contrail cirrus that dissipates within a few hours. Hence, fundamental differences exist in the time dynamics of the climate impacts of $CO_2$ emissions and contrail cirrus. This temporal difference in climate impacts is critical when evaluating mitigation options and policy instruments for the aviation sector.

Currently analyzed mitigation strategies for reducing contrail cirrus forcing include a reduction of soot emissions through (a) the use

[1]Division of Physical Resource Theory, Department of Space, Earth and Environment, Chalmers University of Technology, Gothenburg, Sweden. [2]Department of Economics, School of Business, Economics and Law, University of Gothenburg, Gothenburg, Sweden. [3]Department of Civil and Environmental Engineering, Imperial College London, London, UK. ✉e-mail: daniel.johansson@chalmers.se

of fuels with a lower aromatic content than present fossil jet fuel[11–13], (b) the use of engines emitting less soot[14] and (c) through rerouting of flights so as to avoid contrail forming regions[15–17]. To make informed decisions regarding rerouting of flights, fuel choices, engine development and other potential mitigation measures, the climate impacts of different forcers must be made comparable and the uncertainties of the impacts of different mitigation options need to be assessed. Hence, an understanding of how to value the climate impacts of a short-term and uncertain forcer such as contrail cirrus vis-a-vis the climate impact of the long-term forcer $CO_2$ is critical for the design of cost-efficient mitigation strategies. Further, for mitigation strategies such as flight planning and rerouting, it is critical to also understand the heterogeneity of the atmospheric conditions leading to contrail formation.

The climate impacts of different forcers are typically assessed using emission metrics[18]. These can either be physics-based, e.g., Global Warming Potential (GWP) or be based on economic approaches, e.g., the social cost[19,20]. The GWP is by far the most widely used emission metric[21]. Its value for a given climate forcer is given by the integrated change in ERF over a chosen time horizon following an emission pulse of that forcer, divided by the corresponding estimate for an emission pulse of $CO_2$[21].

A social cost-based approach offers an alternative[19,20]. The social cost of a climate forcer is defined as the net present value of future damages caused by an emission pulse of that forcer. This value depends on the discount rate, which gradually reduces the present value cost of impacts occurring further into the future, the damage function, economic growth, and the temperature pathway considered.

In relation to GWP, the social cost-based approach considers additional dimensions that have theoretical appeal for characterizing the marginal climate impact of emissions. (1) It is based on integrated economic damage rather than integrated radiative forcing. Damages are further down the cause-effect chain compared to radiative forcing implying a larger socioeconomic relevance, but also larger uncertainty[22]. (2) It adopts the use of discounting instead of an integration time horizon. Discounting has a stronger welfare-theoretical appeal than the use of an arbitrary time horizon. (3) It takes into account that the marginal impact of one additional emission unit is larger for higher temperature increases. This is consistent with the climate impact assessment literature[23,24].

The aim of this paper is four-fold:
1. Develop a methodology to estimate the social cost of $CO_2$ (SCC) and contrail cirrus (SC-contrail) in a consistent framework,
2. estimate SCC and SC-contrail, considering uncertainty along a number of modeling dimensions, as well as contrail cirrus forcing heterogeneity,
3. compare the ratio of SC-contrail to SCC with the GWP, and
4. examine how the SCC, the SC-contrail, and the role of risk aversion to uncertain climate impacts affect the balance between using more fuel and avoiding contrails.

The analysis in the paper goes beyond previous research on the social cost of aviation $CO_2$ and contrail cirrus[25–27] in several ways. First, we analyze how updated assumptions on the damage function, future climate pathways and the discount rate affect the social cost of carbon and contrail cirrus; unlike earlier aviation-focused literature, we use a Ramsey discounting approach. Rather than a fully probabilistic approach[25,27], we examine the impact of a set of specific modeling assumptions to make their impacts on the social cost estimates explicit. Second, we introduce the concept of Social Cost of Energy Forcing (SCEF) as an approach to estimate SC-contrail. Third, we consider the impact of contrails on the $CO_2$ concentration through climate-carbon cycle feedback. Forth, we estimate the social costs not only on a globally averaged level, but also on a per flight basis by analyzing the heterogeneity in the SC-contrail and SCC for half a million flights over the North Atlantic region.

To estimate the SCC and SC-contrail (see also Supplementary Note 1), we (i) update the integrated assessment model, Dynamic Integrated Climate-Economy (DICE)[20,28], and (ii) use an assessment of the energy forcing for contrail cirrus based on output from the Contrail Cirrus Prediction (CoCiP) model[6,29].

We analyze three sets of cases for the social costs: (i) three different assumptions for the discount rate using a Ramsey discounting approach—a low rate (about 2%/year)[30], a medium rate (about 2.4%/year)[31], and a high rate (about 4.3%/year)[28], (ii) three different future global emissions and temperature pathways —a 3 °C stabilization pathway representing a case where the world continues with existing policies[32], a 2 °C stabilization pathway representing conditional Nationally Determined Contributions (NDC) and existing net zero pledges[32,33] and a pathway largely consistent with the Paris Agreement[34] that peaks at 1.8 °C and then declines to 1.5 °C warming by 2100 (see also Supplementary Note 2); and (iii) two damage functions— the damage function by Nordhaus[28] and the damage function by Howard & Sterner[23].

We treat contrail forcing uncertainty using a probabilistic approach by: (i) an estimate of systematic (non-weather related) uncertainties in the energy forcing calculations based on a literature assessment; (ii) weather-related variability for specific flights using contrail-cirrus energy forcing estimated for flight-specific conditions with ten ensemble members of the ERA5 reanalysis dataset[35]; (iii) uncertainty in the efficacy of contrail cirrus energy forcing, i.e., the ratio of the climate sensitivity parameter for contrail cirrus to the climate sensitivity parameter for $CO_2$, characterized with a probability density function estimated from the literature. The same efficacy is applied for both global contrail cirrus forcing estimates as well as for the flight-specific forcing estimates.

With our modified version of DICE (M-DICE) we estimate the SCC and SCEF. SCEF is the social cost per GJ energy forcing for an ideal short-lived forcer with an efficacy equal to one and an atmospheric lifetime of less than a year. These estimates of the social costs are subsequently applied to $CO_2$ emissions from global aviation and efficacy adjusted contrail cirrus forcing estimates[6], as well as to the $CO_2$ emission and efficacy adjusted contrail cirrus forcing for individual flights over the North Atlantic region[29]. The aviation $CO_2$ emissions and contrail forcing is based on data representative for 2019[6,29] with additional uncertainty estimates[2], while the social cost estimates are for 2020. For more details on models, assumptions and uncertainty estimates, see the Methods section.

## Results
### Social cost of carbon and social cost of short-lived forcers
The social cost of $CO_2$ is strongly dependent on the discount rate and the damage function, and also, but to a lesser extent, the future temperature pathway[20,31,36], see Fig. 1. Lower discount rates, higher damage costs and higher future temperature pathways increase the SCC. When using the damage function in Howard & Sterner[23] and a low discount rate, the SCC is 910, 1200, and 1800 US$/ton $CO_2$, for the 1.5 °C, 2 °C, and 3 °C temperature pathways, respectively. For the medium discount rate, the corresponding estimates are 400, 510, and 710 US$/ton $CO_2$, while for the high discount rate case, the estimates are 84, 93, and 110 US$/ton $CO_2$. Hence, it can be seen that when the discount rate is high, the impact of the temperature pathway on the SCC is rather low. When using the damage function from Nordhaus[28], the social costs are slightly more than a factor of 3 lower for each case, respectively. Hence, spanning between 26 and 510 US$/ton $CO_2$ for the cases discussed above.

The estimated SCC values can be compared  with selected estimates in the literature[20,28,31,36,37]. Especially, our case with a low and medium discount rate and the use of the damage function by Howard & Sterner[23] gives relatively high SCC values. One reason that we obtain higher SCC values than in Hänsel et al.[31] and Azar et al.[20], even when

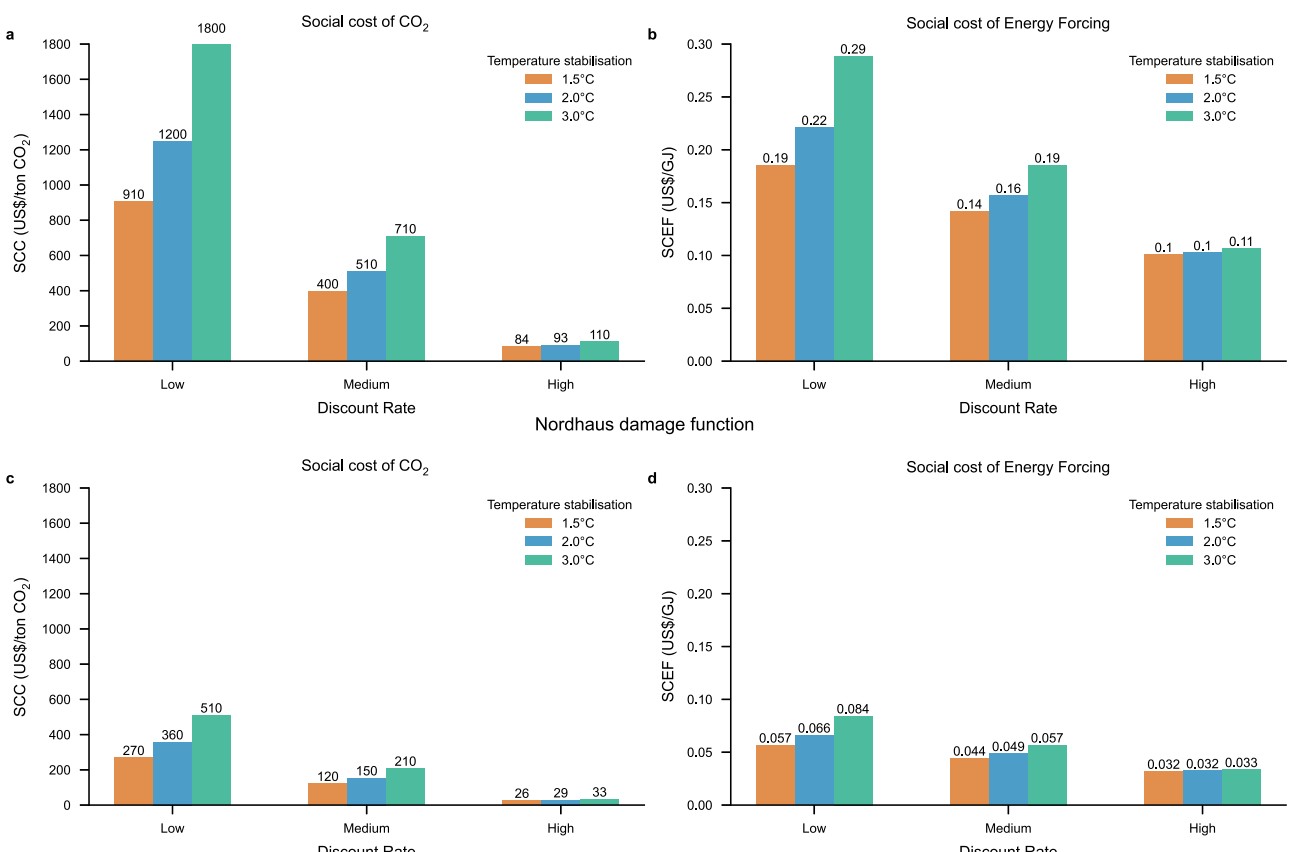

**Fig. 1 | SCC and SCEF in 2020 for three different climate stabilization levels (1.5, 2, and 3 °C), three different discount rates (low, medium, and high), and for two different damage functions. a** shows the SCC (in $/ton CO₂) based on the damage function by Howard & Sterner[23], **b** shows the SCEF (in $/GJ) based on the damage function by Howard & Sterner[23], **c** shows the SCC based on the damage function by Nordhaus[28] and **d** shows the SCEF based on the damage function by Nordhaus[28].

using the same discount rate and damage function, is the use of exogenous temperature pathways where the long-term temperature pathways in our analysis stabilize between 1.5 °C to 3 °C above the pre-industrial level, while in Hänsel et al.[31] and Azar et al.[20] the temperature pathway is optimized and drops to considerably lower levels beyond 2100. The long-term temperature pathway has an impact on the social cost of long-lived greenhouse gases since the marginal damage caused by greenhouse gas emissions increases with increasing temperature when using a quadratic damage function and this effect is particularly pronounced when the discount rate is low but hardly noticeable when the discount rate is high (see Fig. 1 and also Supplementary Note 4 where social cost estimates are also presented for two additional temperature pathways).

Furthermore, we find higher SCC estimates than Rennert et al.[36] when using the damage function by Howard & Sterner[23] but lower when using the damage function by Nordhaus[28]—despite using similar discount rates. A reason for this is that the main case damage function used by Rennert et al.[36] generates damages that are intermediate between our two cases for similar increases in the global mean surface temperature. There are also other differences in the modeling approach, for instance, Rennert et al.[36] use a certainty equivalent approach, while we use a deterministic approach to calculate SCC values.

When using the damage function in Howard & Sterner[23] and a low discount rate, the SCEF is 0.19, 0.22, and 0.29 US$/GJ, for the 1.5 °C, 2 °C, and 3 °C temperature pathways, respectively. For the medium discount rate case, the corresponding estimates are 0.14, 0.16, and

0.19 US$/GJ, while for the high discount rate case the estimates are 0.10 to 0.11 US$/GJ. Like for SCC, the SCEF, when using the damage function in Nordhaus[28] is slightly more than a factor of three lower than when using the damage function in Howard & Sterner[23].

Of the modeling assumptions analyzed, those pertaining to the damage function have the strongest influence on the SCEF, while assumptions concerning the discount rate and temperature pathway have a much smaller effect for the SCEF than the SCC. The reason for this is that the generic forcer is short-lived. Still, it is noteworthy that even though the lifetime of the generic short-lived forcer is short the SCEF still depends, as can be seen in Fig. 1, on both the discount rate and the temperature path. This is explained by the thermal inertia of the oceans and climate-carbon cycle feedbacks[38,39].

### Global social cost of aviation CO2 and contrails

In 2019, global aviation CO₂ emissions were 885 MtCO₂, and contrail-cirrus energy forcing was 999 EJ yr⁻¹ (RF = 62.1 mW m⁻²)[6]. Using our main estimate of the contrail cirrus efficacy of 0.42 (see the "Methods" section for discussion about efficacy assumptions) we get an efficacy-adjusted energy forcing of 420 EJ/yr, with an estimated 5% to 95% interval of 136 to 1410 EJ/yr, corresponding to an efficacy adjusted RF of 26.1 W/m², and a 5% to 95% interval of 8.5 to 88 mW/m² (see the Methods section for uncertainty assumptions). Based on the emissions and efficacy-adjusted energy forcing levels, the total global social cost of aviation CO₂ and contrail cirrus is estimated (see Fig. 2). The estimated global total social cost of aviation CO₂ varies between US$23 billion/yr and US$1600 billion/yr depending on the discount rate, the

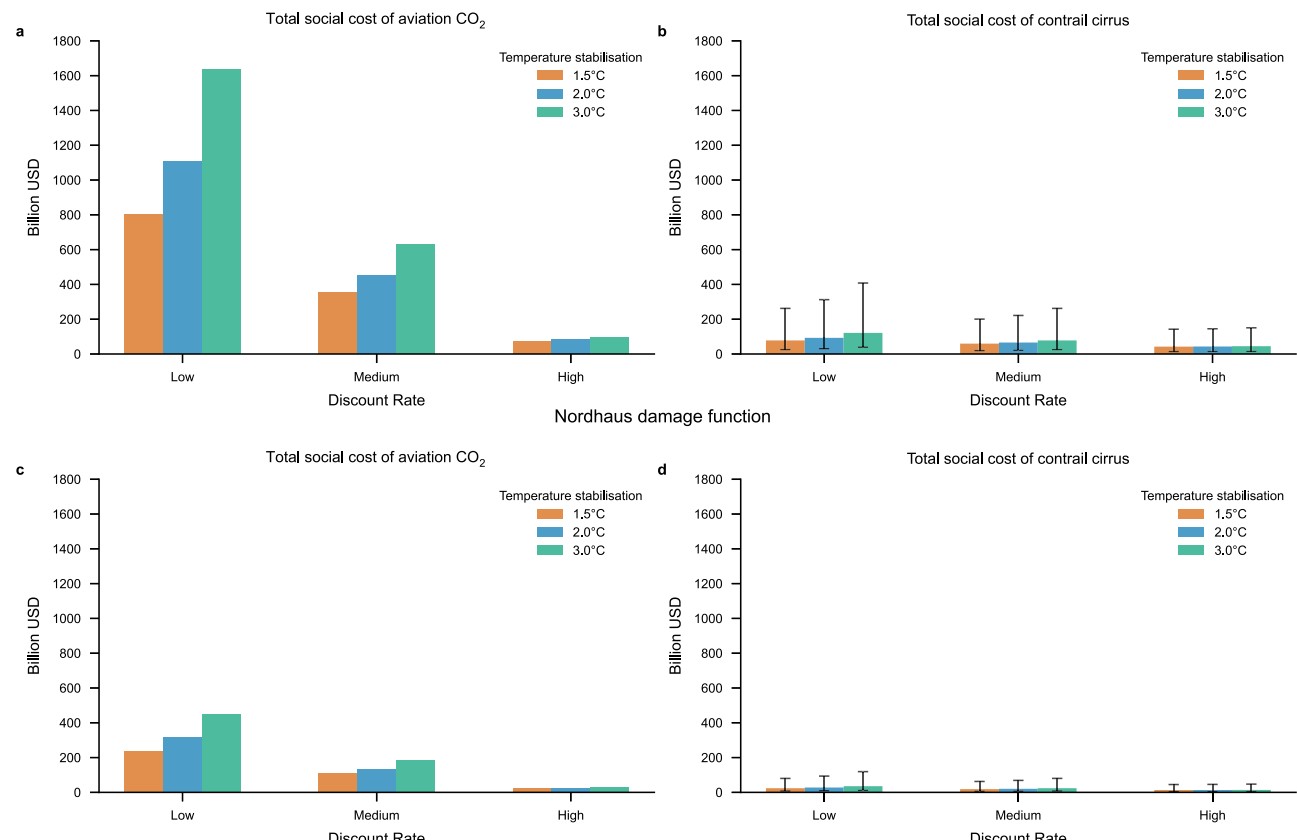

Fig. 2 | **Social costs for aviation $CO_2$ and social costs for contrail cirrus for the emissions and forcing levels in year 2019 based on social cost estimates for 2020.** Results are based on three different temperature stabilization pathways (1.5, 2, and 3 °C), three different discount rates (low, medium and high), for two different damage functions and for a 90% uncertainty interval for the efficacy-adjusted contrail cirrus energy forcing. Uncertainty bars for the social cost of contrail cirrus are due to uncertainties in the forcing and efficacy estimates. **a** Shows the aviation $CO_2$ related social costs based on the damage function by Howard & Sterner[23], **b** shows the aviation contrail cirrus social costs based on the damage function by Howard & Sterner[23], **c** shows the aviation $CO_2$ related social costs based on the damage function by Nordhaus[28] and **d** shows the aviation contrail cirrus social costs based on the damage function by Nordhaus[28].

temperature pathway, and the damage function, while the global total social cost of contrail cirrus forcing varies between US$4.3 billion/yr and US$410 billion/yr depending on the same factors as well as on the uncertainty of the efficacy adjusted energy forcing. Hence, the global social cost of aviation $CO_2$ is in general higher than the global social cost of contrail cirrus in most cases analyzed. The exception is for the high discount rate case when efficacy-adjusted contrail cirrus energy forcing is at least 740 EJ/yr, corresponding to an efficacy-adjusted RF level of at least 46 mW/m².

To put the social cost estimates presented above into context, the global revenue of commercial airlines in 2019 was US$838 billion, with total fuel expenses amounting to US$190 billion[40].

## Ratio of social costs and comparison with Global Warming Potentials

The ratio of the time-integrated global contrail cirrus ERF to the corresponding ERF from aviation $CO_2$ emissions gives the GWP values commonly presented for aviation[2,18]. Here we present the ratio of the global social costs of contrail cirrus to the global social costs of aviation $CO_2$ emissions as a complement to GWP estimates, see Fig. 3. In a central case, with a medium discount rate and a medium temperature path (2 °C stabilization), we estimate this metric to be approximately 0.15.

The uncertainty range is considerable, as the ratio depends on the discount rate, the temperature pathway, and the assumed

efficacy-adjusted contrail cirrus forcing (Fig. 3), with the largest source of uncertainty stemming from the efficacy-adjusted contrail energy forcing. On the other hand, the ratio is essentially independent of the choice of the two damage functions used in the paper since the damage proportionality constant cancels out when taking the ratio between the social costs, and for that reason, only the estimates based on the damage function in Howard & Sterner[23] are shown in Fig. 3.

As the discount rate increases, the social cost ratio of contrail cirrus to $CO_2$ also increases (ceteris paribus), because the SCC decreases relatively more than the SC-contrail. Furthermore, the higher the future temperature pathway is, the lower the ratio (ceteris paribus), again because it increases the SCC more than the SC-contrail. The sensitivity to the discount rate is stronger than to the temperature pathway.

For a given contrail cirrus forcing level, the lowest ratio is obtained with a relatively high temperature pathway (stabilization at 3 °C) and a low discount rate, while the highest ratio is obtained with a relatively low temperature pathway (stabilization at 1.5 °C) and a high discount rate. For the low discount rate case the ratio in our main case is between 0.075 and 0.097, for the medium discount rate case the ratio is between 0.12 and 0.17, while in the high discount rate case the ratio is 0.46 and 0.57. Hence, the case with the highest ratio is a factor 7.7 larger than the case with the smallest ratio (for a given assumption on the efficacy adjusted contrail cirrus energy forcing.

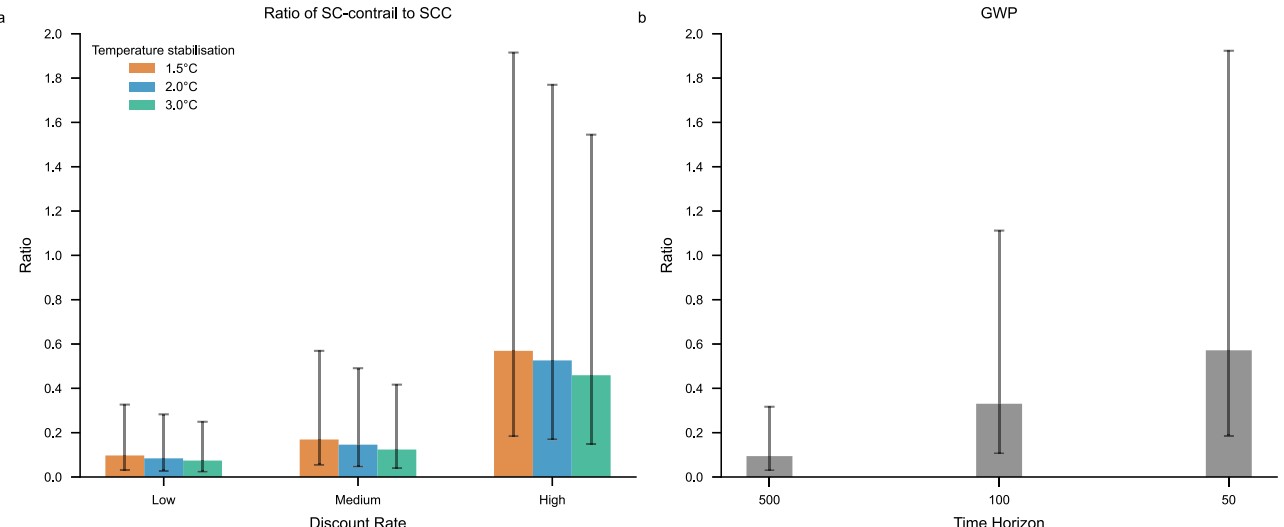

**Fig. 3 | Ratio of social cost of contrail cirrus to social cost of CO₂ and GWP values for aviation contrail cirrus. a** Shows the ratio of social cost of contrail cirrus to social cost of aviation CO₂ emissions under the different assumptions for the temperature stabilization level (1.5, 2 and 3 °C) and the discount rate (low, medium and high), and **b** shows the GWP values for aviation contrail cirrus to aviation CO₂ emissions estimated with time horizons 50, 100 and 500 years. The uncertainty bars reflect the 90% uncertainty interval in the estimate of the efficacy-adjusted EF of global annual average contrail cirrus.

The social cost ratios can be compared to GWP estimates, formally being the ratio of the time-integrated contrail cirrus effective radiative forcing divided by the corresponding time-integrated forcing of the $CO_2$ emissions[21], while we here instead use the efficacy-adjusted radiative forcing. Hence, GWP is a measure of the relative impact of the time-integrated efficacy adjusted contrail cirrus forcing to the time-integrated forcing per unit emission of $CO_2$. The impulse response function and radiative efficiency for the atmospheric concentration impact of $CO_2$ emissions are the same as in IPCC AR6. We use three different integration time horizons.

We find that the best estimate GWP for contrail cirrus (measured on a per mass unit $CO_2$ emission basis) is 0.57 for a 50-year time horizon, 0.33 for a 100-year time horizon, and 0.094 for a 500-year time horizon. As illustrated in Fig. 3, there are similarities between how GWP depends on the time horizon and how the ratio of social costs depends on the discount rate. This holds even though GWP has its roots in physical science, and the ratio of social cost has its roots in economics[20,26,41–44]. The fundamental reason for this similarity is that while GWP is based on integration of efficacy-adjusted RF, the social cost is based on an integration of the temperature response to a pulse emission and the fact that there is a close link between the integrated efficacy adjusted RF and the integrated temperature[45]. Furthermore, the time horizon in the GWP measure plays a role that is closely related to the discount rate, or more specifically the inverse of the effective discount rate, where the effective discount rate is defined as the discount rate minus the economic growth rate[20,26].

Consequently, using a short time horizon, for example 20 years, is not consistent with any of the discount rates used here. Further, a prescriptive approach to discounting based on the median view among economists and philosophers[46,47] suggests that GWP-100 gives too high a relative value for contrail cirrus in a welfare-maximizing context.

As discussed in the introduction, metrics based on social costs are further down the cause-effect chain compared to radiative forcing. This increases the relevance of the metric, although such an approach will entail larger uncertainties[22,48]. Still, it is important to keep in mind that the uncertainties in the relative metrics are not necessarily much larger for social cost-based approaches since the numerator and denominator covary for the assumptions concerning the discount rate, future climate pathway, and the damage function.

## Application of social cost estimates to North Atlantic flights

Figure 2 shows the total SCC and SC-contrail for the global aviation sector in 2019, and Fig. 3 shows their ratio and the corresponding GWP estimates. However, individual flights may produce orders of magnitudes more contrail forcing per ton fuel used than the global average flight. To capture the heterogeneity and uncertainty of the social cost of contrail cirrus from individual flights, we analyze contrail cirrus EF and $CO_2$ emissions from 477,923 flights over the North Atlantic region during 2019[29].

Based on this dataset we analyze (1) the heterogeneity of the SC-contrail to the SCC for each flight, i.e. using model output on contrail forcing and $CO_2$ emission caused by each flight, and (2) make a preliminary analysis of what proportion of these flights would benefit (from a climate economic perspective) from being rerouted to avoid the formation of contrail cirrus, given a specific level of fuel penalty.

The results in the subsequent sections are presented for the three different discount rates, for the 2 °C temperature stabilization pathways, and for the damage function in Howard & Sterner[23]. As discussed above, the choice of proportionality coefficient in the damage function has essentially no impact on the climate trade-off between $CO_2$ emissions and contrail cirrus, and the temperature pathway has only a small impact (Fig. 3).

Figure 4 shows the ratio of the flight-specific social cost of contrail cirrus to the social cost of carbon dioxide, with flights grouped into bins representing 2% of the total number of flights, ordered from the lowest to the highest ratio based on the main run (using the contrail EF per flight generated with weather input from control member of the ERA5 10 ensemble members and assuming an efficacy of 0.42). The flight specific SC-contrail in the main case is calculated as the SCEF multiplied by the efficacy-adjusted energy forcing caused by the flight along its path over the North Atlantic region generated by the control run in the different ERA5 ensemble. The uncertainty in the ratios is obtained by calculating the SC-contrail for each flight using ten different ERA5 ensembles members in which contrail cirrus forcing estimates are further scaled by the probabilistic forcing and efficacy multiplier.

About 14% of the flights in the main run cause contrail cirrus with negative energy forcing, i.e., they are cooling the planet, as shown in the left panels in Fig. 4. About 38% of the flights cause persistent contrail cirrus with positive energy forcing, i.e., warming, as shown in

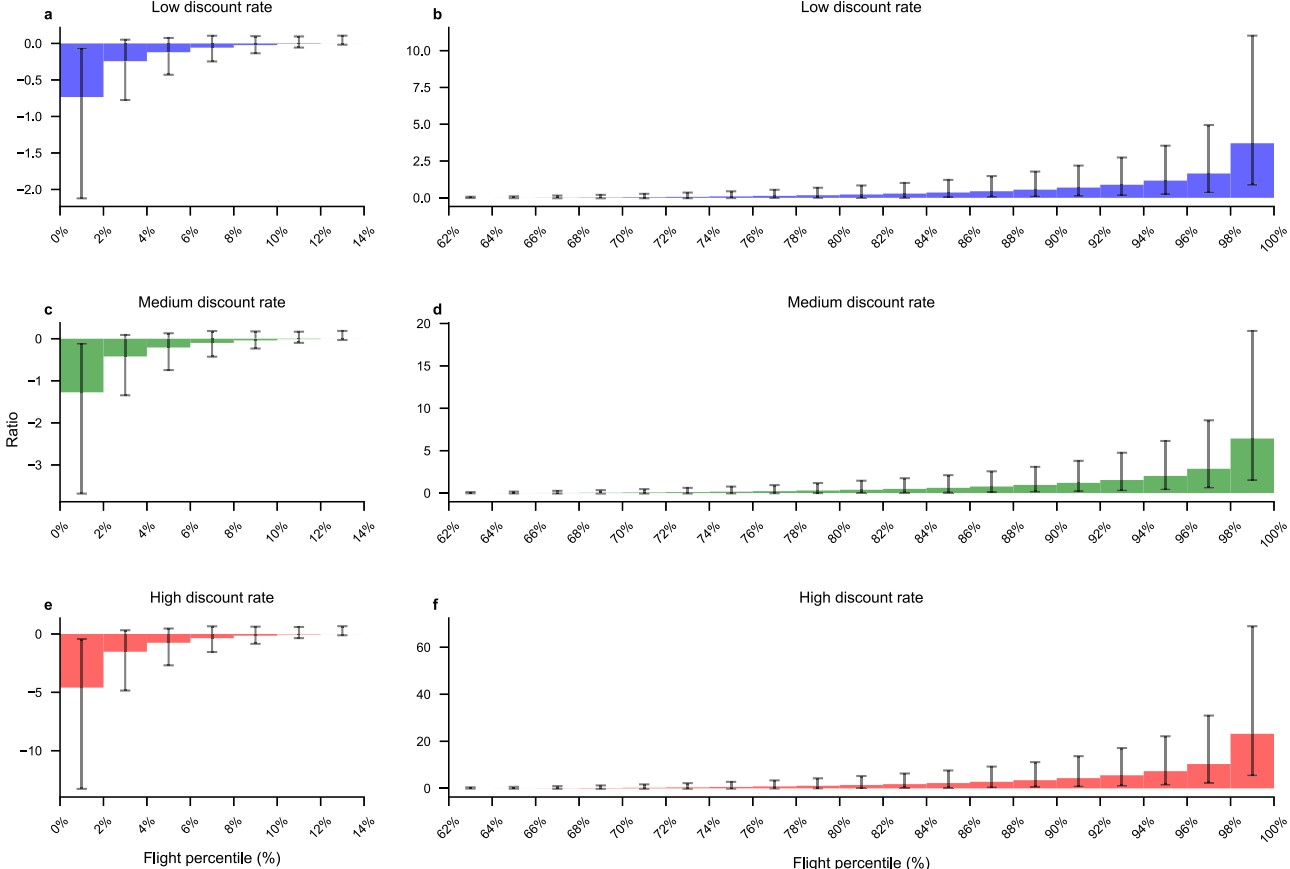

**Fig. 4 | The ratio of SC-contrail caused by a flight to the SCC caused by the same flight ordered from lowest to highest ratio for 477 923 flights over the North Atlantic in the year 2019 and then divided into bins representing 2% of the total number of flights.** The bars represent the social cost ratio in the main case, while the whiskers represent the average of the 90% uncertainty interval within the bin. **a**, **c**, **e** show the ratio for the cooling contrails, and **b**, **d**, **f** show the ratio the warming contrails. The social cost ratios for the flights not shown (12% to 66%) are close to zero since few and/or weak contrails are formed for these flights. **a, b** are for the low discount rate case, **c, d** for the medium discount rate case and **e, f** for the high discount rate case. All results are based on the 2 °C stabilization pathways using the damage function by Howard & Sterner[23] (the ratio is virtually identical when using the damage function by Nordhaus[28]).

the right panels in Fig. 4. The remaining flights do not generate any persistent contrails at all in the main run. If the ratio is −1 the economic valuation of the climate impact of the cooling contrails cancels the economic valuation of the warming impact caused by the $CO_2$ emissions along the flight path. Hence, if the ratio is below −1 the flight leads to a net climate benefit in economic terms. Flights generating cooling contrails, causing a net climate benefit, happens rarely, in the order of a few percent of the flights, depending on the contrail cirrus efficacy of the EF and the discount rate. Higher discount rates make such cases more common, with a greater share of flights having a ratio below −1. Conversely, a ratio above 1 indicates that the economic impact of contrail cirrus is greater than that of $CO_2$ emissions along the flight path. This occurs in the main contrail cirrus forcing case for approximately 6% to 22% of flights, with the frequency increasing at higher discount rates.

We now turn to analyzing the potential benefits of rerouting and carry out an analysis for each of the 477,923 flights. If we take an illustrative approach and assume that rerouting can be done with a fuel use penalty of 1% per flight, consistent with Borella et al.[18], then all flights for which the social cost ratio is above 0.01 would lead to a net reduction in climate impact. We also assume here that rerouting with certainty prevents the formation of contrails and we do not include the cost of the fuel, only the social cost of the associated $CO_2$ emissions. By analyzing the data behind Fig. 4, we find that 34% of the flights in the main case would benefit from rerouting to avoid contrails (33%, 34%, and 35% for the low, medium, and high discount rates, respectively). If

the fuel penalty were as large as 5% per flight, it would be beneficial from a climate impact perspective to reroute fewer flights, (28%, 30% and 33% respectively), i.e. corresponding to when the social cost ratio is 0.05 or higher in Fig. 4. Notably, the discount rate only plays a minor role for the result.

If we instead assume that policy makers are risk-averse, i.e., that they have a preference for certainty that a rerouting choice actually leads to a climate benefit, and assume that rerouting should only be undertaken when the probability that there is a net positive climate benefit is higher than 95%. When using such an assumption, between 21% and 24% of the flights ought to be redirected in the 1% fuel penalty case, and between 16% and 22% in the 5% fuel penalty case (depending on the discount rate).

In Fig. 5 we show the estimated cumulative gross climate benefits (in economic terms) obtained for flight rerouting based on the main estimate of efficacy adjusted contrail EF for each flight, the 90% uncertainty interval efficacy adjusted contrail EF for each flight, and for different assumptions on the discount rate and the damage function. The cumulative gross social costs of emitting more $CO_2$, based on a fuel penalty of either 1% or 5%, are also presented in Fig. 5.

The climate benefits of contrail avoidance outweigh the additional social costs of carbon from increased fuel use. Notably, a relatively small share of flights contributes to a large share of the contrail cirrus EF (about 10% of the flights contribute with about 80% of the energy forcing)[29]. This implies that the potential net climate benefit of rerouting these flights would be large even if rerouting fuel penalties

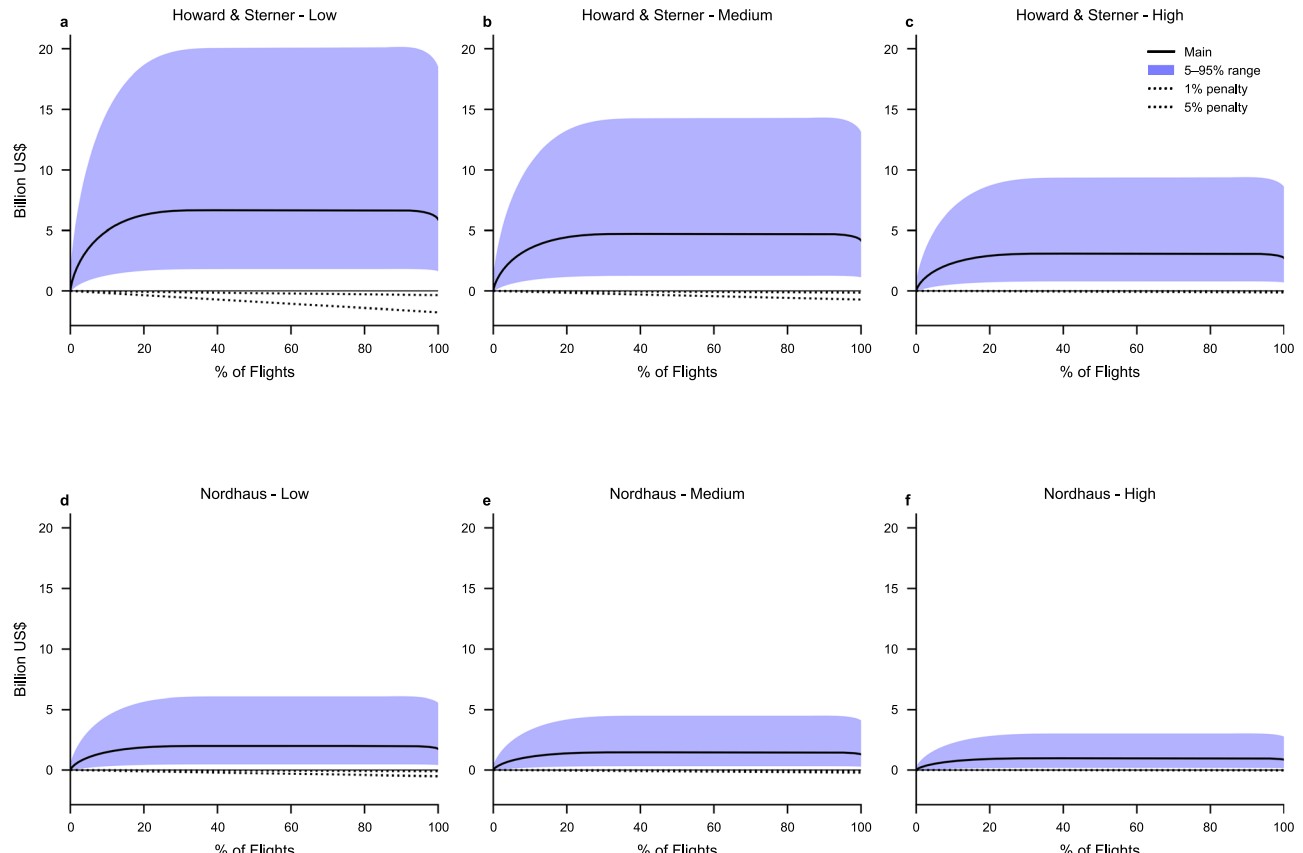

**Fig. 5 | Cumulative climate benefits of avoided contrails for specific flights for the main case together with corresponding 90% uncertainty intervals, ordered from the highest to the lowest EF per flight in the main case.** The cumulative benefits are calculated by progressively adding the climate benefits of flights with decreasing main case EF. **a**–**c** display cumulative benefits using the Howard & Sterner[23] damage function, while **d**–**f** display cumulative benefits using the Nordhaus[28] damage function. **a**, **d** are based on the low discount rate, **b**, **e** are based on the medium discount rate, and **c**, **f** are based on the high discount rate.

are on the order of 5% per flight. The magnitude of the benefit depends strongly on the uncertainty in the contrail EF and efficacy, the choice of damage function, and the discount rate.

## Discussion

In this study, we analyze the social costs of aviation's $CO_2$ emissions and contrail cirrus. The analysis is presented both in aggregate and at the individual flight level for nearly half a million flights across the North Atlantic region. Combining a range of assumptions on discount rates, temperature pathways, and damage functions, as well as uncertainties and heterogeneities in the warming impact of contrail cirrus, we provide a broad range of estimates of social costs. The analysis highlights the complexities behind short-lived, uncertain, and heterogenous climate forcers like contrail cirrus alongside the more predictable and long-lived $CO_2$ emissions.

We estimate the global average ratio of the social cost of contrail cirrus (SC-contrail) to the social cost of $CO_2$ emissions (SCC) to be approximately 0.15 with a medium discount rate and a 2 °C temperature target. This ratio provides a measure of the relative impact of contrail cirrus compared to $CO_2$, though, as expected, it is strongly dependent on assumptions. By varying the discount rate, temperature pathways, and assumptions about efficacy adjusted contrail cirrus forcing a broad range of potential ratios, spanning from about 0.02 to 2, is found. The highest ratios are associated with a high discount rate, a low temperature pathway, and an efficacy-adjusted contrail cirrus forcing level in the upper end of the plausible range, while the opposite assumptions give a low ratio.

It is important to note that while a higher discount rate raises the SC-contrail to SCC ratio, it also reduces the absolute values of both SC-

contrail and SCC. This implies that high SC-contrail to SCC ratios alone do not justify an intensified focus on contrail mitigation from an economic standpoint. In comparison to the Global Warming Potential (GWP) metric, a social cost-based approach offers a method that goes further down the cause-effect chain, yielding a more relevant number, but with larger uncertainties. However, our estimates for the social cost ratio of contrail cirrus to $CO_2$ emissions are similar to GWP values when the integration time horizons are set to be consistent with the discount rate assumption.

The analysis for specific flights in the North Atlantic region shows large variability in contrail cirrus impacts. Approximately 14% of flights analyzed produce a negative energy forcing (i.e., cooling the planet), 48% produce no persistent contrails, and the remaining 38% contribute positive (warming) forcing through contrail cirrus formation. For flights with warming contrail cirrus, we observe a wide range in relative climate impact, with a subset of flights generating contrail impacts an order of magnitude greater than the corresponding $CO_2$ emissions from the same flight. Looking in isolation at these flights, there appears to be an important opportunity for policy. The share of these flights where the flight-specific SC-contrail is an order of magnitude higher than the flight-specific SCC depends on the discount rate and the uncertainty in contrail EF and efficacy. This variability and heterogeneity in contrail impact across flights point to the critical role of targeted strategies in contrail mitigation.

Our analysis can also be used for offering preliminary insights into rerouting as a mitigation strategy. We find that when using the main estimate for contrail cirrus forcing for specific flights and if the additional fuel penalty remains below 1%, rerouting would be climate beneficial for about 35% of flights, being more than 90% of all flights

expected to generate net worming contrails. For a fuel penalty as high as 5%, the flights that would be beneficial to reroute decrease to about 30%. The numbers presented here may not be representative for other regions than the North Atlantic region. However, the approach adopted in the paper is generalizable to other regions.

With this said, the high degree of uncertainty surrounding contrail cirrus formation, persistence, and radiative forcing necessitates a cautious approach to any mitigation policy. Our analysis incorporates uncertainty concerning weather and radiation based on a reanalysis approach, as well as an assumption of structural uncertainties in contrail cirrus forcing using probabilistic scaling on EF output from CoCiP and probabilistic assumptions on the efficacy. By considering 1000 efficacy-adjusted energy forcing estimates per flight, the results indicate that higher certainty requirements for net climate benefit of rerouting (e.g., 95%) decrease the proportion of flights beneficial to reroute for a given level of fuel penalty. However, the uncertainty analysis of rerouting should be interpreted with caution, as substantial scientific knowledge gaps remain in characterizing the warming impacts of individual contrails and their associated uncertainties.

The insights provided in this study offer valuable perspectives for EU's ongoing efforts to integrate non-$CO_2$ aviation effects into climate mitigation regulations and other related policy efforts. Starting in 2025, aircraft operators will be required to report estimated non-$CO_2$ impacts of their operations, and by 2028, the European Commission aims to propose legislation to mitigate non-$CO_2$ aviation effects[49]. Understanding the implications of the climate impact of different forcers, their heterogeneity, and uncertainty, is essential for designing cost-efficient policy instruments. For example, applying a simple adjustment factor (constant across all flights) to $CO_2$ emissions to account for non-$CO_2$ effects would lead to inefficiencies. By acknowledging the variability in warming impacts, policies could be structured to economically incentivize reductions in strongly warming contrails while paying less attention to weakly warming or cooling contrails. Regulatory (non-economic) policies may also be an option.

However, substantial uncertainties and complexities currently make it difficult to forecast as well as ex-post estimate contrail formation and their warming impacts[50,51]. Effective policy instruments need accurate forecasts of contrail longevity and radiative properties to enable informed mitigation planning. Additionally, any rerouting efforts must comply with air traffic control constraints and safety concerns, adding further complexity to an already challenging strategy. Potential initial policy focus could be on minimizing contrail cirrus effects by targeting high-impact flight slots and routes, such as afternoon and nighttime flights during winter along contrail-prone routes.

Finally, for contrail cirrus mitigation initiatives, it is critical to determine not only the effective radiative forcing from the expected contrails, but (1) how to value climate impacts over time and (2) how to weigh certain versus uncertain impacts of potential mitigation strategies. These decisions are inherently value-laden. Science alone cannot determine an appropriate discount rate (or cutoff time horizon when estimating GWPs) nor establish the level of certainty on net climate impact required to classify a mitigation measure as effective. The greater the certainty we require in reducing climate impacts, the more we should prioritize abatement options with quantifiable climate benefits. Similarly, the more we value long-term climate outcomes, the more emphasis we should place on reducing emissions of long-lived greenhouse gases.

## Methods
The calculations of social costs of contrail cirrus and $CO_2$ are based on combining output from a modified version of DICE[28], which we call M-DICE[20,31], together with output from the CoCiP model[6,29]. Below, we explain and motivate how DICE has been modified, how we use output from CoCiP[6,29] together with M-DICE output to estimate social costs of contrail cirrus and $CO_2$, and how we deal with uncertainty in the modeling.

### M-DICE
DICE is based on an integrated Ramsey-Koopmans-Cass model of optimal economic growth hard-linked to a climate model[52]. DICE models $CO_2$ emissions and their abatement cost, and the impact of emissions on the carbon cycle, global mean surface temperature, and damages to the economy. The objective is to maximize welfare, defined as the net present value of population multiplied by utility of per capita consumption for the world over 2015–2515. The model in its standard form seeks the optimal balance between abatement costs and climate damages, so that the marginal abatement cost is, at each time period, equal to the marginal damages per ton of emissions. However, the model can also be set to meet an exogenous global temperature target at the lowest possible cost. In this study, we use the latter approach and estimate the Social Cost of Carbon (SCC) and Social Cost of Energy Forcing (SCEF) for paths towards various, exogenously set, temperature targets.

In Azar et al.[20], based on Hänsel et al.[31], DICE 2016R2 was revised in the following way: (1) the damage function was recalibrated[23], (2) the parameters governing the discount rate were based on a survey among economic experts[46], (3) methane and nitrous oxide emissions and abatement cost functions were added to the model, (4) atmospheric gas cycles ($CO_2$, $CH_4$, and $N_2O$) were based on FaIR 2.0.0 covering the non-linearity in the atmospheric lifetime of the gases[53], and (5) the energy balance model used for calculating the global mean surface temperature was based on a calibration that emulates large-scale climate system models[54].

In this study, M-DICE is used to calculate the SCC and the SCEF under various assumptions. The SCEF can subsequently be used to estimate the SC-contrail under varying conditions. The social cost estimates are for the year 2020, presented in US$–2020 prices.

The social costs are estimated by running the model in two steps (see also Supplementary Note 1):

1. In the first step, we generate emission pathways consistent with five different global temperature targets by using M-DICE, of which three pathways are used for the main paper, and two additional pathways are used for results presented in Supplementary Note 4. In this step, the damages are not considered in the optimization. M-DICE is instead solved with a constraint on the maximum temperature change, and the emission pathway generated is the one that maximizes welfare given that constraint. Hence, this pathway is consistent with the least cost abatement pathway for the specific climate target.

2. M-DICE is then subsequently run with the damage function included and the abatement levels for the emissions of each gas being fixed from step 1. In this step, the shadow price of a unit emission (being the social cost per unit emission) in 2020 is estimated for each forcer ($CO_2$ and a marginal increase in energy forcing affecting the annual average radiative forcing) for the different temperature pathways, the different discount rates, and the different damage functions. The SCEF is subsequently used to estimate the SC-contrail by multiplying its value with global estimates of contrail cirrus EF or by the contrail cirrus EF generated by each specific flight (see Eqs. 3–6 below).

The results from IAMs in general, and especially simple IAMs with a modeling time horizon stretching several hundred years into the future, should always be interpreted with care. There are many uncertainties, and in this study, we emphasize key uncertainties impacting the SCC and SCEF. We explicitly analyze the sensitivity concerning the choice of discount rate, future climate pathway, the damage function calibration, contrail cirrus forcing uncertainty, and heterogeneity. However, there are uncertainties and potential feedbacks within the integrated climate and economic system that we do not account for. The results should be seen as preliminary and subject to revision as knowledge progresses.

## Discounting

The social discount rate in DICE is determined by the "Ramsey rule", i.e.,

$$r = \eta g + \delta \tag{1}$$

where $r$ is the discount rate, $\eta$ the absolute value of the elasticity of marginal utility of consumption, $g$ the annual growth rate of per capita consumption, and $\delta$ the pure rate of time preference (also referred to as the utility discount rate). Some authors have argued that the pure rate of time preference should be set to a low value or even zero, since it determines how current generations value the welfare of future generations. The fact that we may be impatient when it comes to decisions about our own individual lives is not a legitimate reason for discounting the value of future generations[30,47,55,56]. The first term refers to a wealth effect: discounting due to the elasticity of marginal utility of consumption stems from the fact that people tend to derive less additional satisfaction for each additional unit of consumption when their overall consumption is higher.

In principle, the use of discounting implies that future climate impacts are valued progressively less over time. Consequently, the social cost of the forcer $X$ at time zero (in our case, the year 2020), denoted $SC_X$, depends on the discounted future climate impacts and can be expressed as:

$$SC_X = \int_0^\infty \frac{\partial D(t)}{\partial T(t)} \cdot \frac{\partial T(t)}{\partial E_X(t_0)} e^{-rt} dt \tag{2}$$

where $D(t)$ is the climate damages at time $t$, being the number of years after the emission pulse, $T(t)$ the temperature response at time $t$ and $E_X$ represents the emission (or generation) of forcer $X$ at time 0.

In this paper, we will use three representative cases for the discount rate parameters, a "low", a "medium", and "high" case. We set $\eta = 1$ and $\delta = 0.1\%$/yr in the low case following Stern[30]. In the medium case we set $\eta = 1$ and $\delta = 0.5\%$/yr following the "median expert view" in Hänsel et al.[31] and in the high discount case we follow Nordhaus[28] and set $\eta = 1.45$ and $\delta = 1.5\%$/yr. With an average growth rate in per capita consumption of about 1.9%/yr over the 21st century as we get in M-DICE it implies an average discount rate of about 2%/yr in the low discount case, about 2.4%/yr in the medium discount rate case, and about 4.3%/yr in the high discount rate case.

The calibration of $\eta$ and $\delta$ is largely on par with the assumptions used by Rennert et al.[36]. The assumptions used by Rennert et al.[36] imply, if used together with an average growth rate in per capita consumption of about 1.9% per year over the 21st century, a discount rate of 1.9 %/yr in their low case (with $\eta = 1.02$ and $\delta = 0.01\%$/yr), 3.2%%/yr in their main case (with $\eta = 1.42$ and $\delta = 0.5\%$/yr), and 3.8%/yr in their high case (with $\eta = 1.57$ and $\delta = 0.8\%$/yr).

## Damage function

In DICE, there is a damage function that translates a specific temperature level to monetary damage. This damage function is proportional to the global mean surface temperature increase squared and global GDP. The proportionality constant is subject to a lot of uncertainty and debate. Nordhaus[28] has estimated a value for the proportionality constant so that the damage is about 2.1% of global GDP for a temperature increase of 3 °C. In this paper, we use both the damage function by Nordhaus et al.[28] and the results from a meta-study of the available literature by Howard & Sterner[23] who find a damage of 6.7% of global GDP for a temperature increase of 3 °C. Both higher and lower damage estimates are available in the literature[24].

The choice of proportionality constant for the damage function matters for the estimates of the SCC and SC-contrail. Still, the ratio between the two, which measures how serious the climate impacts of

contrail cirrus are compared to $CO_2$ emissions from aviation, is essentially independent of the damage proportionality constant.

We assume that the damage function applies to short-lived forcers such as contrail cirrus. The climate response of short-lived forcers does not follow the same pattern as that of long-lived forcers, but still cause widespread impacts far beyond the local area where the forcing occurs[57–60]. The relative global average temperature response of short-lived climate forcers compared to that of $CO_2$, for equal radiative forcing, is captured by multiplying the climate sensitivity by the efficacy value. Further, the climate impacts of short-lived forcers tend to be more localized, with temperature impacts primarily within the hemisphere where the emissions occur, and especially strong along the zonal direction[58,59,61]. Given that the majority of contrail cirrus forms in the northern hemisphere, and the fact that majority of the global population and the global economic activity is in the northern hemisphere it is possible that using the global average temperature response of the short-lived forcer, a damage function based on global mean surface temperature and global estimates of GDP underestimates the social cost of short-lived contrail cirrus.

Related arguments apply for the contrails formed by flights in the North Atlantic region (more specifically in the region 10°W to 50°W and 40°N to 75°N). Although these contrails may be formed far from the majority of economic activities, the impact of these forcers likely has an impact over large parts of the Northern hemisphere, especially in the zonal direction.

Lund et al.[62] analyzes the distribution of temperature impacts of regional aviation forces. The North Atlantic region (which is placed in our paper within 50°W and 10°W and 40°N and 75°N) largely falls within their "EUR" emission source region[62]. Contrails sourced in the EUR region show a powerful temperature impact in the latitude bands 28°N and 90°N, and are in their paper about twice as large as the global average temperature impact of contrails sourced in EUR region[62].

## Estimating SC-contrail

Energy forcing, for an individual contrail (denoted $EF_i$ for contrail cirrus $i$), is the top of the atmosphere (TOA) local radiative forcing $RF'_i(t)$, i.e., the forcing per unit area given in unit $W/m^2$, multiplied by the area of the contrail cirrus, where $H_i(t)$ is width and $L_i(t)$ length, and integrated over its total lifetime $\tau_{contrail,i}$.

$$EF_i = \int_0^{\tau_{contrail,i}} RF'_i(t) \cdot H_i(t) \cdot L_i(t) dt, \tag{3}$$

From M-DICE, we can estimate the shadow price of efficacy-adjusted RF, $SC_{eRF}$, being the social cost of $e \cdot RF$ where $e$ is the efficacy. Hence, the marginal welfare loss of one unit global annual average efficacy-adjusted radiative forcing expressed in monetary units [US $\$ \, W^{-1} \, m^2$] is defined as

$$SC_{eRF,t} = \frac{\partial W / \partial (e \cdot RF_t)}{\partial W / C_t}, \tag{4}$$

where $W$ denotes net present value welfare, $e \cdot RF_t$ denotes the $e \cdot RF$ in year $t$, and $C_t$ denotes consumption in the year $t$. The concept of $SC_{eRF}$ has previously been used[63,64]. In this paper, we take this approach one step further and estimate the SC of Energy Forcing [US$/J] by calculating:

$$SC_{eEF} = \frac{SC_{eRF}}{\tau_{year} \cdot A_{earth}}, \tag{5}$$

where $\tau_{year}$ is the number of seconds per year and $A_{earth}$ area of Earth. Based on the SC of efficacy-adjusted Energy Forcing (eEF), we can

estimate the SC-contrail cirrus $SC_{cc,i}$, where $i$ refers to each flight or each contrail (where one flight may cause several contrails along its path) as follows:

$$SC_{cc,i} = SC_{eEF} \cdot e_c \cdot EF_i \qquad (6)$$

where $e_c$ is the is the efficacy for contrail cirrus.

The SCC is defined as

$$SCC_t = \frac{\partial W / \partial E_{CO_2,t}}{\partial W / C_t}, \qquad (7)$$

in which $E_{CO_2,t}$ denotes the $CO_2$ emissions in year $t$.

The efficacy-adjusted GWP for contrails is estimated as:

$$GWP_{cc,H} = \frac{e_c \cdot EF_i}{\int_o^H \gamma_{CO_2} \cdot IRF_{CO_2}(t) \cdot \tau_{year} \cdot A_{earth} dt}, \qquad (8)$$

where $\gamma_{CO_2}$ is the effective radiative efficiency of $CO_2$, $IRF_{CO_2}$ the impulse response function for $CO_2$ emissions, and finally $H$ is the integration time horizon. When estimating GWP, we use the $\gamma_{CO_2}$ and $IRF_{CO_2}$ used in IPCC AR6[21].

**Aviation $CO_2$ emissions and contrail cirrus forcing assumptions**

Global $CO_2$ emissions from aviation in 2019 were estimated to 885 MtCO$_2$[6]. Teoh et al. (6) also estimate the radiative forcing (RF) from contrail cirrus for the same year as 62.1 mW/m². In comparison, Lee et al.[2] provide a best estimate of contrail cirrus RF in 2018 to 114 mW/m², with a 90% uncertainty interval ranging from 33 to 189 mW/m² based on a literature assessment. The estimate by Teoh et al.[6] is lower than that of Lee et al.[2], due to model differences and as a result of the fact that Teoh et al.[6] use of a flight route dataset that better reflects existing flight paths.

To enable a probabilistic approach to estimating the social cost of contrails, we fit a gamma distribution with a mode equal to the global RF estimate by Teoh et al.[6] and a 90% interval near the 90% interval estimated by Lee et al.[2] The parameters of the gamma distribution are estimated by constraining the mode to 62.1 mW/m² while minimizing the squared deviation at the 5th and 95th percentiles to those estimated by Lee at al.[2]. The procedure generates a shape parameter equal to 3.15, and a scale parameter equal to 0.466. The 90% interval for our fitted distribution spans from 26 to 189 mW/m², with an average of 91 mW/m².

The contrail cirrus forcing for specific flights is very heterogeneous. To illustrate the implications of this, we analyze 477,923 flights passing over Shanwick and Gander Oceanic Air Traffic Control Area in the North Atlantic region, which approximately corresponds to the area within longitudes −50°W and −10°W and latitudes 40°N and 75°N, during 2019[29]. Of these 477,923 flights, about 52% cause persistent contrails[29].

Estimating the uncertainty in energy forcing for the contrail cirrus generated by specific flights is challenging, see Supplementary Note 3 for further discussions. There is only little guidance on this in the scientific literature, with an exception being Platt et al.[65]. The approach taken here is crude and is performed to generate preliminary insights that should be refined as scientific knowledge progresses. The approach taken here to incorporate uncertainty in energy forcing for the contrail cirrus generated by specific flights is based on the following approach:

1. The estimates of the flight-specific contrail cirrus energy forcing are based on weather and radiation conditions sourced from the ERA5 reanalysis of the European Centre for Medium-Range Weather Forecasts (ECMWF) [33]. To account for weather uncertainties, we make use of the 10-member ensemble, which

has a 0.5° × 0.5° horizontal resolution, 37 pressure levels, and a 3-h temporal resolution[35]. This ensemble is used in Teoh et al.[29] for estimating the variability in contrail properties and forcing. We use their output on contrail energy forcing for the 477,923 flights for each of these 10 ensemble members to characterize the uncertainty, i.e., 10 contrail cirrus EF estimates per flight. For our main assumption for contrail cirrus forcing per flight, we use the output from CoCiP based on the control run of the ensemble. Note that ERA5 has been reported to have various systematic challenges for contrail modelling[65,66]. To reduce some of these limitations, Teoh et al.[29] apply a methodology to correct for biases in relative humidity.

2. The 10 ensemble members account for weather and radiation variability and how that affects the contrail characteristics formed by each flight. However, when aggregated to annual forcing levels for all 477,923 flights the differences between the 10 different ensembles become small, since the impact of weather variability cancels out[29,65]. To account for model parameterization uncertainties as well as model structural uncertainties (i.e., uncertainties in equations and modelling approach) in the forcing estimates by CoCiP, the contrail cirrus EF for each flight obtained in each of the 10 ensemble members is scaled by an uncertainty distribution with the same relative form as the one used for the RF uncertainty at the global level. This is a simple approach, but goes beyond parametric uncertainty estimates[65]. However, our uncertainty distributions for flight-specific contrail cirrus EF are comparable to, but wider than, those of Platt et al.[65], see Supplementary Fig. 7.

Efficacy-adjusted RF, as well as Effective Radiative Forcing (ERF), is lower than RF for contrails. The reason for ERF being lower than RF depends on rapid adjustments of natural clouds following contrail formation, i.e., the formation of contrails and contrail cirrus tends to reduce the natural cirrus cloud cover due to a limited atmospheric water budget[2,67]. Further, additional differences in slow feedbacks relevant for the climate sensitivity value of equally large ERF of contrail cirrus and of $CO_2$ may cause the efficacy adjusted RF for contrails to be even lower than the ERF[68]. Several studies have attempted to estimate the ratio of ERF to RF and the efficacy of contrail cirrus. Lee et al.[2] assume a best estimate for the ERF to RF ratio of 0.42. This number is based on earlier studies, including Rap et al.[69], who found an efficacy of 0.59; Ponater et al.[70], who reported an efficacy of 0.31; and Bickel et al.[67], who found an ERF to RF ratio of 0.35 with a 90% uncertainty interval between 0.23 and 0.51.

Additionally, in a later study, Bickel[68] finds a contrail efficacy of 0.21. For our probabilistic analysis, we assume a normal distribution for the efficacy on RF (i.e., an efficacy value including the impact of differences for both fast and slow feedbacks for contrail cirrus vis-à-vis $CO_2$), with a mean of 0.42 and a 90% uncertainty interval from 0.2 to 0.64, covering the range of values found in the literature. The use of a normal distribution implies that there is a very small probability that the efficacy is zero or negative (less than 0.1% for the assumed distribution). It is very unlikely, but not physically impossible, that the efficacy for contrail cirrus could be negative[71].

Using an RF estimate of 62.1 mW/m² and an efficacy of 0.42 as our main estimate, we get the efficacy-adjusted RF to be 26.1 mW/m², as in Teoh et al.[6] When we compound the uncertainty distributions for RF and efficacy, the 90% range becomes 8.5 mW/m² to 87.8 mW/m², with an average of 38.2 mW/m². This can be compared to published estimates of ERF: Gettelman et al.[72] report a value of $62 \pm 59$ mW/m² (the uncertainty interval corresponding to two standard deviations), and Lee et al.[2] estimate 57.4 mW/m², with a 90% interval from 17 to 98 mW/m². Hence, our estimate is somewhat lower than that of Gettleman et al.[72] and Lee et al.[2], largely because we set the mode of the gamma distribution to

Teoh et al.[2,6,72] and that we include relatively low efficacy values in our distribution.

The uncertainty in the efficacy of individual contrails is assumed to be the same as on the global level. This is likely an underestimation of the efficacy uncertainty for individual contrails since this value may depend on a range of local conditions. However, for lack of better scientific knowledge, we maintain this estimate.

When assessing the distribution of the social cost of individual flights, we have 10 realizations based on the different ensembles runs, and in addition, we take 100 random draws from the multiplication of the EF scaling distribution and the efficacy distribution. Each random draw on the combined EF scaling and efficacy distributions is multiplied by the EF per flight for each of the 10 ensemble runs, leading to 1000 possible estimates for the-SC contrails for each flight.

## Data availability
The input data and the source data for the figures have been deposited in Zenodo with https://doi.org/10.5281/zenodo.14507267[73].

## Code availability
The source code for generating figures has been deposited a Zenodo with https://doi.org/10.5281/zenodo.14507267[73]. Source code of M-DICE is available from the corresponding author upon reasonable request.

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

## Acknowledgements
Funding has been provided by VINNOVA grant number 2023-01286, Chalmers Area of Advance Transport, Chalmers Area of Advance Energy, and the Familjen Kamprads Stiftelse project 20230142.

## Author contributions
D.J.A.J. initiated the research and performed the computational analysis. C.A., D.J.A.J. and T.S. wrote the original draft, while S.P., R.T. and M.E.J.S. contributed to further review and editing. R.T. and M.E.J.S. provided data on $CO_2$ emissions and contrail energy forcing for North Atlantic flights.

## Funding

## Competing interests
The authors declare no competing interests.
