## [Peer Review file · Nature Communications]

The social costs of aviation CO₂ and contrail cirrus

Corresponding Author: Dr Daniel Johansson

Version 0:

Reviewer comments:

Reviewer #1

(Remarks to the Author)

The paper “The Social Cost of Aviation: Comparing Contrail Cirrus and CO₂” concerns itself with quantifying the social cost of carbon and contrails for global aviation. The paper is interesting and timely as it addresses current policy needs in the context of monitoring and potentially including aviation non-CO₂ emissions in the EU-ETS. This is important given the high significance of aviation non-CO₂ emissions, potentially making up 50% of the aviation climate footprint. An updated view on the relative importance of CO₂ and non-CO₂ impacts provides much needed guidance, especially on the high significance of discount rates in determining such ratios. The paper derives these results by running the integrated assessment model M-DICE and by including an estimate of the impacts of contrails which are estimated separately via the contrail model COCIP and a shadow price for ERF from DICE. While I generally like the approach, I have some concerns and questions as outlined below.

1. Uncertainty of contrail impacts

The authors, in the abstract, state that this analysis is needed to provide insights on how to balance “certain” CO₂ impacts with “uncertain and heterogeneous” contrail impacts. I think this is a very important statement. In the paper, the authors assess, in detail, the heterogeneity of contrail impacts on a flight-by-flight basis (see Figure 4). Yet, I do not see assessments of uncertainty (neither in global averages, nor in flight-by-flight estimates). Given the ranges of uncertainty in contrail impacts (e.g., Lee et al., 2021), these uncertainties can have substantial impacts on the balance between CO₂ and non-CO₂ impacts – which would need to be considered both in global average estimates, but also (and very importantly) in the evaluation of each contrail’s impact plotted in Figure 4. I do not see a sufficiently detailed discussion or acknowledgement in the paper. Such an uncertainty analysis is probably exceptionally important for decisionmaking – which the paper is explicitly trying to support. For example, if policies are meant to reduce non-CO₂ impacts and if they do so by operational contrail avoidance (i.e., by flying around regions where persistent contrails form), we need to understand the balance of CO₂ and non-CO₂ impacts under this uncertainty. For the paper to provide sound insights, I think uncertainty in the SC-contrails needs to be quantified. Ideally, the authors find a way to address this explicitly, or I highly recommend caveating the paper accordingly. Some of the implications probably then require softening, for example the statements in lines 263-269, but also in the Discussion Section.

2. Central metrics used in the analysis

The analysis initially focuses on contrail impacts per unit CO₂ emitted (Section 2.1). While I understand that the metrics are based on global averages, I fundamentally struggle with such a SC-contrail metric. The CO₂ emission per se is not the driver of contrail formation. There are even trade-offs between CO₂ and contrail formation in operational contrail avoidance. As such, I think the CO₂ standardization can easily be misunderstood. I understand that the authors use this normalization to assess the relative magnitude of CO₂ to contrail impacts. To solve this conundrum, I wonder if it would be better to compare the total cost of CO₂ emissions vs. the total cost of contrails (as presented in the discussion session) – just to avoid misunderstandings.

I very much enjoyed the discussion of metrics in Section 2.3 – but I feel that the discussion comes too late in the paper – especially the interpretation of different normalization metrics (lines 211-216) could be more prominent (and probably slightly more in-depth).

3. Novelty

While the paper addresses an important topic, I note that the idea of quantifying aviation non-CO2 impacts through monetary metrics and comparing them to CO2 impacts is not new (e.g., Dorbian et al. 2011 and Grobler et al., 2019). From those papers, the sensitivity of the impact estimates to discount rates is well-known. I recommend that this paper acknowledges this prior work and provides a comparison of results. More importantly, I think it would be important for the authors to work out what exactly the additional contribution of this new paper is, as compared to such prior studies. There is some very limited discussion, for example in lines 93 to 98, but I worry that this is insufficient to put the paper into context.

4. Additional questions and comments

[Line 43]: I recommend avoiding expressions like “highly” or “large” which do not mean much. Quantify if possible. There are a few other places in the paper where this is the case. Please take a look.

[Lines 78f]: “This is beneficial since damage is closer to what really matters than radiative forcing” – I see what you are getting at. The sentence seems very cryptic though and there might be a benefit in being more explicit / clear about this point.

[Section 2.1]: I recommend also comparing the SC-CO2 to Rennert et al. (2022) in Nature. It seems to me that your SC-CO2 are high – and I am not sure I fully understand why. Please discuss in more detail.

[Line 173]: I recommend not to talk about a “link” between GWP and the ratio of social cost – that technically does not seem quite accurate to me. Fundamentally, there are similar concepts at work really.

[Section 2.4]: I recommend caveating what we can and cannot learn from the analysis of the North Atlantic flights.

[Table 1]: I find this Table hard to read and hard to infer meaning from it. It also does not seem quite in line with the text – possibly as the percentile cases are not fundamentally linked to each other? Is it better to just show the ratios and plot a distribution of those?

[Line 229]: “fFigure” (there are a few instances in the paper where you capitalize / do not capitalize “Figure”. You might want to be consistent).

[Line 277]: “Understanding the implications of heterogeneity is crucial for the design of cost-effective policy instruments” – I would challenge this statement because this world is full of uncertainty in which a policy does not have to be “right” on a flight-by-flight basis. This points to my concerns around uncertainty discussed above.

[Lines 323-331] Does this mean that you cannot capture feedbacks between warming of contrails and aviation CO2? It probably doesn't matter too much but would be good to understand.

[Lines 326-331]: I am worried about the integration of ERF into DICE. How exactly is this done?

[Lines 346-350]: I very much appreciate the idea of Ramsey discounting, but I think you might want to provide an initial discount rate in the text for calibration.

Reviewer #2

(Remarks to the Author)

The manuscript “The Social Costs of Aviation: Comparing Contrail Cirrus and CO2” delves into an essential but often overlooked aspect of aviation's climate impact: contrail cirrus. By comparing the social costs of contrail cirrus with those of CO2 emissions, the authors provide valuable insights that could inform policy decisions. However, to enhance the robustness and depth of their analysis, several key areas warrant further exploration. The following comments offer specific suggestions for the authors to consider in their revisions.

Strengths:

Addresses a knowledge gap: This paper illuminates the often overlooked but significant social costs of contrail cirrus, highlighting an important aspect of aviation's climate impact.

Comparative analysis: By comparing the social costs of CO2 and contrail cirrus, the paper provides valuable insights to guide policymakers in reducing aviation's overall climate impact.

Emphasizes variability: The study underscores the varying climate effects of contrail cirrus based on different atmospheric conditions.

Examines influencing factors: The research delves into how discount rates, temperature pathways, and contrail ERF estimates affect social cost calculations.

Policy implications: The discussion includes potential cost-effective strategies for mitigating climate impacts, considering the variability of contrail effects.

Points for Improvement:

Methodology transparency: It would be beneficial to offer more detailed explanations of how the DICE model was updated to include contrail cirrus effects.

Alternative damage functions: Consider assessing the impact of using different damage functions, beyond those of Howard & Sterner (2017), to test the robustness of the results.

High SCC estimates: Provide a clearer rationale for the higher social cost of carbon (SCC) estimates compared to other studies and discuss the implications for policy-making.

Limitations of GWP comparison: While the connection between GWP and social cost ratios is intriguing, acknowledge the limitations of GWP as a purely physical metric.

Expand rerouting analysis: Enhance the discussion on rerouting by including more detailed considerations of route changes, fuel efficiency, and technological improvements.

Policy recommendations: Develop more specific policy recommendations that take into account both the social cost estimates and the variability in contrail cirrus effects.

Additional Considerations:

Ongoing mitigation research: Briefly mention current research efforts aimed at reducing contrail formation through operational or technological solutions.

Limitations of the social cost approach: Discuss potential limitations, such as uncertainties inherent in economic models and damage functions.

Overall, the manuscript provides significant insights into the social costs of contrail cirrus and their implications for aviation policy. By addressing the suggested improvements, the authors can enhance the paper's strength and its contribution to the field.

Strengthening the Analysis:

Discount Rate Sensitivity: The paper explores the use of different discount rates, but it would be beneficial to delve deeper into the rationale for including a low rate, such as 2%. Is this rate a realistic representation of societal preferences for future generations? Furthermore, consider the broader implications of using a wider range of discount rates, especially those reflecting a stronger emphasis on future impacts.

Temperature Pathway Dependence: The analysis indicates that the social cost of contrail cirrus varies depending on future temperature pathways. To enhance this aspect, it would be worthwhile to explore whether this dependency becomes more pronounced under higher temperature scenarios. Investigate if more extreme temperature pathways significantly affect the social cost estimates.

Contrail ERF Uncertainty: While the paper highlights uncertainty by using two different estimates for contrail ERF, this could be further quantified using a probabilistic approach. By doing so, the analysis would provide a clearer picture of how this uncertainty impacts the social cost estimates and what it means for policy decisions.

Distance-Based Social Costs: The introduction of distance-based social costs for both CO₂ and contrail cirrus is a useful metric. However, it may not fully capture the complexities of climate impact that vary with flight paths and atmospheric conditions. It would be helpful to explore these limitations and consider how a more refined metric could offer a better understanding of the climate impacts.

Addressing Potential Issues:

Global Average vs. Specific Flight Analysis: The contrast between global average social costs and the significant heterogeneity observed in specific flights is an important finding. However, the limitations of relying on global averages for policy decisions should be acknowledged. It's crucial to highlight the importance of considering this heterogeneity to develop more effective mitigation strategies.

Rerouting Feasibility: The idea of rerouting flights to avoid contrail formation is intriguing, but it's important to address the practical challenges and limitations. Discuss factors such as air traffic control constraints, potential economic costs, and the feasibility of alternative mitigation strategies to provide a more comprehensive view.

Additional Considerations:

Equity Concerns: It's important to briefly touch upon the equity concerns related to aviation's climate impact, especially how it disproportionately affects developing countries. This adds a valuable socio-economic perspective to the discussion.

Reviewer #3

(Remarks to the Author)

The paper explores the social costs associated with aviation emissions, focusing specifically on the climate impacts of contrail cirrus and CO₂. It aims to provide a framework for estimating and comparing the social costs of these two

contributors to climate change. The authors employ the Dynamic Integrated Climate Economy (DICE) model, considering different discount rates and future climate pathways. They analyze the sensitivity of the social cost of contrail cirrus and CO₂ to these factors and highlight the significant variability in contrail cirrus costs due to specific meteorological conditions.

Overall, it is my opinion that this paper may not yet be suitable for acceptance. My primary concerns are outlined below:

1. Lack of a clear definition of the contribution of this paper:

The paper lacks a concise description of its main contribution. While the findings are clear, it is necessary to provide specific results, especially regarding the significant variation in social costs under different discount rates. The definition and significance of the discount rate, a crucial aspect of the paper, are not sufficiently explained. Clear definitions and explanations are essential for readers to understand the paper's core concepts.

2. Lack of specificity in the importance of comparing the social cost of contrail cirrus and CO₂:

The purpose and significance of comparing the social costs of contrail cirrus and CO₂ emissions need to be more explicitly specified. The paper should clearly state why this topic is important and elaborate on the relevance of this comparison within the context of aviation emissions.

3. Need for clarification on assumptions about global average conditions and the North Atlantic Region:

The authors mention assumptions related to global average conditions and the comparison with the North Atlantic Region. However, these assumptions require further explanation to provide a comprehensive understanding of the study. Clarifying these assumptions would enhance the transparency and applicability of the research.

4. Insufficient explanation regarding the innovative contribution and regional comparisons:

The contribution of the paper, as described between lines 93 and 98, appears to be limited to a typical usage in a particular region. To establish a stronger innovative point, it would be beneficial to compare different regions rather than solely focusing on the North Atlantic Region. This would better illustrate the heterogeneity and improve the paper's overall impact.

5. Need for more detailed explanations of result numbers:

The paper should provide more comprehensive explanations of the result numbers presented. For instance, between lines 169 and 172, where numbers such as 0.57, 0.33, and 0.094 are listed, it is crucial to clarify the meaning and implications of these values. Providing additional context and interpretation of the result numbers will enhance the reader's understanding.

6. Lack of explanation regarding the influence of discount rate:

Between lines 239 and 240, the paper briefly mentions the influence of the discount rate without providing sufficient details. It is important to elaborate on why the discount rate is influential and its significance within the context of the study. A more thorough exploration of the discount rate's impact would enhance the paper's credibility and provide a clearer understanding of its implications.

7. Request for a more comprehensive explanation of the calculation methods:

The paper should incorporate a detailed explanation of the calculation methods within the main body of the article rather than confining them to the appendix. Presenting the calculation methods in the context of the study would facilitate a better understanding of the research and its findings.

8. Typographical errors:

Several typographical errors are present in the manuscript. These include "CO₂and" in line 15, "Ctemperature" in line 148, and a missing "." at the end of a sentence in line 289. Careful proofreading and correction of these errors are necessary to enhance the overall quality of the paper.

9. Concerns regarding the treatment of contrail formation and overestimation:

The authors acknowledge the dependence of the social cost of contrail cirrus on specific meteorological conditions. However, it seems that the paper treats the airline itself as the contrail, which may lead to an overestimate of the influence of contrail cirrus, especially considering the aim of comparing the impacts of contrail and CO₂. Clarifying the approach to contrail formation and its representation in the study would address this concern.

10. Simplified model and limitations:

While the use of the DICE model is a common practice in integrated assessment modeling, it is important to acknowledge the limitations it introduces. The simplified representation may overlook important factors and feedback loops, potentially resulting in biased or incomplete social cost estimates. Recognizing this limitation is crucial to avoid overreliance on the model's outputs when formulating policies and making decisions.

Version 1:

Reviewer comments:

Reviewer #1

(Remarks to the Author)

The paper "The Social Cost of Aviation: Comparing Contrail Cirrus and CO₂" concerns itself with quantifying the social cost of carbon and contrails. I am reviewing the second version of the paper. I remain convinced that the paper is interesting and timely. I recognize that the authors have spent considerable time re-structuring the paper and adding additional analyses, including (but not limited to) to address many of my earlier comments. Unfortunately, it is my overall impression that the edits were rushed. The paper now feels incohesive and hard to follow. I strongly recommend that the authors go through the paper with great care to clean up the logic and writing to make the paper accessible. Additional pointers and questions to underpin this overall impression are given in the following:

1. Uncertainty of contrail impacts

The handling of uncertainty in the paper confuses me. It feels like uncertainty analysis was added around the edges of a

modeling framework, which wasn't built to deal with uncertainty. It feels like there is no cohesive approach for addressing uncertainty (but rather piecemeal considerations here and there). Here are several points which I got hung up in this context (and which led me to the overall impression):

- The authors now model uncertainty in contrail formation, persistence and impacts which is done in a probabilistic framework. The damage function uncertainty and temperature pathway uncertainty are dealt with via a sensitivity analyses. They also consider discount rate (I'd argue that discount rate isn't even subject to uncertainty in a strict sense). The analyses are all in different places and it is quite hard to maintain an overall perspective of the results range given the different methods and approaches. In addition, the overall approach for addressing uncertainty isn't described in a cohesive manner in the paper (this became clear to me when re-reading the last 3 paragraphs of the Introduction).
- I appreciate the efforts to model meteorological uncertainties in contrails as well as the efficacy discussion and related uncertainty. However, the framework seems very simplified and might be incomplete. For example, I can't quite see where uncertainty in contrail properties (ice crystals shapes etc.) would be considered.
- In the individual flight model, the authors assume the weather model adjustment in addition to the high-level EF adjustment, which seems to be designed to match the form of the adjustment for the global framework. Isn't that inconsistent (in the sense that the different models might already be capturing the meteorological uncertainties?)? Or am I missing something here?
- Shouldn't the uncertainties be discussed in a more upfront manner, e.g.: include them in Figure 1, 4 and 5 or in the discussion (e.g., see lines 451 ff in the marked up version)?
- In Section 2.2, the authors state a total RF number. The methods section then describes how they get to a distribution of that number. That seems inconsistent. How do these relate?

2. Flight-by-flight analysis

The authors add a flight-by-flight contrail analysis. While I generally appreciate that, they then draw conclusions on the benefits of contrail mitigation based on a fixed fuel burn number. I do not think that this is a valid way of looking at these results, given that we know that the fuel burn penalties for contrail avoidance vary among flights.

3. Global metrics

I wonder whether the approach of calculating a cost of contrail RF via a global shadow price can be valid. The DICE model determines shadow prices of CO₂ and short-term forcers. In contrast to CO₂, contrails are not well mixed. There is evidence of regionalized sensitivities. I agree that this is very hard to capture, but wouldn't it be good to discuss this at the very least? It would raise the question how valid it is to apply global cost metrics to the North Atlantic case (which I continue to feel somewhat uncomfortable with).

4. Novelty

I continue to think that the idea of quantifying aviation non-CO₂ impacts through monetary metrics and comparing them to CO₂ impacts is not new (e.g., Dorbian et al. 2011 and Grobler et al., 2019). I am still missing a clear upfront novelty statement relative to existing work. Some language has been added, but still not acknowledging the aviation-specific work in detail.

5. Additional questions and comments (if line numbers are given, they reflect line numbers in the marked up version of the paper)

- I may have missed it, but somewhere in the paper it might be good to state that this is all done for quantifying the impacts of a flight today (or in a specific year).
- [Line 157] What's the source for the Paris pathway? It seems to be missing.
- In comparisons, the authors use unspecific language like "higher" or "lower", without stating how much higher and lower values are. This makes it harder to read and understand the such comparisons [specifically cumbersome in Section 2.1]. Quantify if possible.
- The introduction talks about contrail mitigation quite a bit. I propose to scale this back as this is not the focus of the paper.
- [Lines 186ff]: Note that there are different reference styles (with author in text or just footnote – I'd prefer the former).
- [Lines 192-197]: Isn't the text repetitive?
- [Lines 256ff]: I recently saw a new publication by Prakash et al. (<https://pubs.rsc.org/en/content/articlehtml/2024/se/d4se00419a>) which also includes total climate costs of aviation. The paper seems to have updates from earlier work by that group (Grobler et al). You may want to look at their numbers and compare them here.
- [Line 680ff] The explanation on how contrails are included in DICE seems short and very hard to understand. In my view, this requires more discussion, as it is one of the key points in the method.
- [Lines 752ff & 797ff]: I propose adding more explanation which distribution functions are assumed and why. In its current form, the discussion seems somewhat arbitrary.

- [Lines 776ff]: Results. Why are they in the methods section?

(Remarks on code availability)

Reviewer #2

(Remarks to the Author)

Suggestions for Minor Improvements (Optional)

While the manuscript is now well-developed, the following minor suggestions may further refine the study:

Consider providing a brief graphical summary or flowchart illustrating the integration of M-DICE and CoCIP models, to enhance accessibility for readers less familiar with modeling frameworks.

Expand slightly on the policy implications of high SCC values under low discount rate scenarios, especially for long-term climate policies.

(Remarks on code availability)

Reviewer #3

(Remarks to the Author)

I appreciate the authors' thorough and thoughtful revisions in response to my comments. They have significantly improved the clarity and precision of their manuscript, particularly in defining the study's contribution, elaborating on the role of the discount rate, and refining explanations related to methodology and assumptions.

The authors have effectively addressed my concerns by:

- Clearly articulating the study's main contributions and methodological innovations.
- Providing a well-defined explanation of the discount rate and its impact on the analysis.
- Strengthening the justification for comparing the social cost of contrail cirrus and CO₂ emissions.
- Enhancing discussions regarding regional assumptions and model limitations.
- Offering clearer explanations of key numerical results and their implications.

Additionally, the corrections to typographical errors and the improvements in methodological descriptions make the paper more precise and accessible. Given these revisions, I now find the manuscript suitable for publication in its current form.

(Remarks on code availability)

Version 2:

Reviewer comments:

Reviewer #1

(Remarks to the Author)

The paper "The Social Cost of Aviation: Comparing Contrail Cirrus and CO₂" concerns itself with quantifying the social cost of carbon and contrails. I am reviewing the third version of the paper. I very much appreciate the time and effort which the authors spent to improve the paper and respond to my earlier comments. The last round of edits very much improved the readability of the paper and addressed my concerns. I think, this is a very insightful and well-rounded paper now – I wholeheartedly recommend it for publication.

Just a few small pointers which, in my mind, can be addressed as a final manuscript is prepared for typesetting:

1 – In the Introduction, there are a couple of subsequent paragraphs which enumerate points, almost in bullet form (Lines 88-128). It would be nice if the writing could be revised a little to make this flow a bit better.

2 – Introduction, Line 91. The authors say that uncertainty analysis is a "focus" of the paper. I think that's still too much to say (and the rebuttal letter seems to agree with that assessment). I propose softening the language maybe something along the lines of "estimate SCC and SC-contrails, considering uncertainty along a number of modeling dimensions, as well as heterogeneity".

3 – Line 191: Revise sentence. Word order seems confused.

4 – Method, Lines 626-637. I would recommend adding one sentence which caveats that ERA5 ensemble analysis would likely not capture any systematic challenges in the underlying data (which is a well-documented concern)

(Remarks on code availability)

Response to reviewers comments

The comments made by the referees are presented in black, our responses to the comments are presented in red, and citations from our manuscript are presented in red italics.

Reviewer #1:

The paper “The Social Cost of Aviation: Comparing Contrail Cirrus and CO₂” concerns itself with quantifying the social cost of carbon and contrails for global aviation. The paper is interesting and timely as it addresses current policy needs in the context of monitoring and potentially including aviation non-CO₂ emissions in the EU-ETS. This is important given the high significance of aviation non-CO₂ emissions, potentially making up 50% of the aviation climate footprint. An updated view on the relative importance of CO₂ and non-CO₂ impacts provides much need guidance, especially on the high significance of discount rates in determining such ratios. The paper derives these results by running the integrated assessment model M-DICE and by including an estimate of the impacts of contrails which are estimated separately via the contrail model COCIP and a shadow price for ERF from DICE. While I generally like the approach, I have some concerns and questions as outlined below.

Author reply: Thank you for the summary of our manuscript.

1. Uncertainty of contrail impacts

The authors, in the abstract, state that this analysis is needed to provide insights on how to balance “certain” CO₂ impacts with “uncertain and heterogenous” contrail impacts. I think this is a very important statement. In the paper, the authors assess, in detail, the heterogeneity of contrail impacts on a flight-by-flight basis (see Figure 4). Yet, I do not see assessments of uncertainty (neither in global averages, nor in flight-by-flight estimates). Given the ranges of uncertainty in contrail impacts (e.g., Lee et al., 2021), these uncertainties can have substantial impacts on the balance between CO₂ and non-CO₂ impacts – which would need to be considered both in global average estimates, but also (and very importantly) in the evaluation of each contrail’s impact plotted in Figure 4. I do not see a sufficiently detailed discussion or acknowledgement in the paper.

Such an uncertainty analysis is probably exceptionally important for decisionmaking – which the paper is explicitly trying to support. For example, if policies are meant to reduce non-CO₂ impacts and if they do so by operational contrail avoidance (i.e., by flying around regions where persistent contrails form), we need to understand the balance of CO₂ and non-CO₂ impacts under this uncertainty. For the paper to provide sound insights, I think uncertainty in the SC-contrails needs to be quantified. Ideally, the authors find a way to address this explicitly, or I highly recommend caveating the paper accordingly. Some of the implications probably then require softening, for example the statements in lines 263-269, but also in the Discussion Section.

Author reply: Thank you for pointing out the importance of considering uncertainties in contrail

forcing, it is indeed a very central issue! We have now updated the paper so that the heterogeneity and uncertainty in the radiative forcing and in the efficacy of contrails are reflected clearly both for the assessment at the global level, as well as for the flight-by-flight assessment. Further, for the flight-by-flight estimate we have added an analysis on the impact of a requirement on the level of certainty (reflecting risk aversion among the policy makers) that a climate benefit is achieved for a hypothetical re-routing case. The entire paper has been updated to focus more on uncertainty, so we are not pointing out any specific sentences in the paper.

2. Central metrics used in the analysis

The analysis initially focuses on contrail impacts per unit CO₂ emitted (Section 2.1). While I understand that the metrics are based on global averages, I fundamentally struggle with such a SC-contrail metric. The CO₂ emission per se is not the driver of contrail formation. There are even trade-offs between CO₂ and contrail formation in operational contrail avoidance. As such, I think the CO₂ standardization can easily be misunderstood. I understand that the authors use this normalization to assess the relative magnitude of CO₂ to contrail impacts. To solve this conundrum, I wonder if it would be better to compare the total cost of CO₂ emissions vs. the total cost of contrails (as presented in the discussion session) – just to avoid misunderstandings.

I very much enjoyed the discussion of metrics in Section 2.3 – but I feel that the discussion comes too late in the paper – especially the interpretation of different normalization metrics (lines 211-216) could be more prominent (and probably slightly more in-depth).

Author reply: In order to avoid the normalisation to emissions of CO₂, we now first present the social cost of carbon and the social cost of energy forcing, i.e., the social cost per GJ energy forcing for a generic forcer with an efficacy equal to 1. This approach is chosen for two reasons: 1. It is consistent with the modelling set up where we first estimate the social cost of CO₂ and social cost of EF using M-DICE and then in a subsequent step estimate the implications for the social cost of contrail cirrus. 2. We believe that this estimate of social cost of energy forcing is useful on its own, for example, it can be applied to issues of operational contrail avoidance where the operator has their own contrail simulation model that estimates the energy forcing for individual contrails and/or flights (it is also a useful number for other short-lived climate forcers besides contrail cirrus such as black carbon). After having introduced the estimate of the social cost of carbon and the social cost of energy forcing, we follow up with an analysis of the total social cost of carbon and total social cost of contrail for the entire global aviation sector. Hence, we follow the suggestion made by the reviewer. This analysis also considers the uncertainty in the efficacy adjusted radiative forcing for contrail cirrus. The metric analysis that follows is built on the ratios of the social costs at the global level. Regarding the order of presentation, we think it is better to present the actual social cost numbers before going into the ratio discussion since such an order aligns more clearly with the actual model approach taken.

Finally, we have now also extended the section where we present social cost ratios and GWP so that it reflects efficacy adjusted forcing uncertainties in the estimates (section 2.3 and figure 3).

3. Novelty

While the paper addresses an important topic, I note that the idea of quantifying aviation non-CO₂ impacts through monetary metrics and comparing them to CO₂ impacts is not new (e.g., Dorbian et al. 2011 and Grobler et al., 2019). From those papers, the sensitivity of the impact estimates to

discount rates is well-known. I recommend that this paper acknowledges this prior work and provides a comparison of results. More importantly, I think it would be important for the authors to work out what exactly the additional contribution of this new paper is, as compared to such prior studies. There is some very limited discussion, for example in lines 93 to 98, but I worry that this is insufficient to put the paper into context.

Author reply: We had a reference to the work by Dorbian et al (2011) in our previous version, as well as to a related study by some of the co-authors of the current paper, i.e. Azar & Johansson (2011), and we have now also added a reference to Grobler et al (2019). To better follow up on the referee's suggestion, we now write the following in section 1: *"The analysis in the paper goes beyond the previous research on the social cost of aviation CO₂ and contrail cirrus in several ways [24, 25, 26]: 1. It contributes with an updated and thorough assessment of the role of the discount rate, the damage function and the future climate pathway for the social cost of carbon and contrail cirrus. 2. It analyzes the heterogeneity in the social cost of contrail cirrus based on half a million flights over the North Atlantic region and 3. It uses the social cost estimates to analyze the implications for rerouting of individual flights when considering the heterogeneity and the uncertainty in contrail cirrus forcing estimate."*

Further, in SM3 we also write the following *"The numbers in presented in table SM1 can also be compared to earlier estimates of the SC-contrail. Dorbian et al (2011) finds a ratio of SC-contrail to SCC between 0.10 and 3.56 depending on discount rate and contrail cirrus forcing assumption. The range obtained using the values in table SM1 is between 0.024 and 2.1. One key reason we find a lower maximum value is that the highest discount rate assumed by Dorbian et al is higher than ours. Grobler et al (2019) estimated the SC-contrail cirrus as US\$ per ton fuel burn, while our estimates in table SM1 are presented as US\$ per ton CO₂ emissions. Converting the estimates by Grobler et al (2019) gives a range between 3.2 and 73 US\$ per ton CO₂ emissions (if assuming 3.16 kg CO₂ per kg fuel burn). Our range is 5.2 to 470 US\$ per ton CO₂ emissions. A key reason for why our upper estimate is so much higher than the upper estimate in Grobler et al (2019) is the use of the H & S damage function."*

4. Additional questions and comments

[Line 43]: I recommend avoiding expressions like "highly" or "large" which do not mean much. Quantify if possible. There are a few other places in the paper where this is the case. Please take a look.

Author reply: Thank you for the observation. We have gone through the document and removed most instances where we had used "highly" or "large" and only kept it in a few instances where it seemed appropriate.

[Lines 78f]: "This is beneficial since damage is closer to what really matters than radiative forcing" – I see what you are getting at. The sentence seems very cryptic though and there might be a benefit in being more explicit / clear about this point.

Author reply: The sentence now reads: "It is based on integrated economic damage rather than integrated radiative forcing. Damages are further down the cause-effect chain compared to radiative forcing implying a larger socioeconomic relevance, but also larger uncertainty [21]."

[Section 2.1]: I recommend also comparing the SC-CO₂ to Rennert et al. (2022) in Nature. It seems to me that your SC-CO₂ are high – and I am not sure I fully understand why. Please discuss in more detail.

Author reply: Yes, our numbers presented in the previous version of manuscript were higher than the numbers by Rennert et al (2022). We have had quite extensive discussions with Rennert in connection with other work. In the new manuscript we present results also with an alternative damage function that leads to lower social estimates. Hence, we now present social estimates based on two different damage function instead of using one damage function. Also we have included a longer discussion of our results on the social cost of carbon in comparison to recent literature. We now write "The estimated SCC values can be compared to estimates in the literature [19, 27, 30, 34, 35]. Especially, our case with a low and medium discount rate and the use of the damage function by Howard & Sterner (2017) [22] gives relatively high SCC values. One reason that we obtain higher SCC values than in Hänsel et al (2020) [30] and Azar et al (2023) [19] is the use of exogenous temperature pathways where the long-term temperature pathways in our analysis stabilize between 1.5°C to 3°C above the pre-industrial level, while in Hänsel et al (2020) [30] and Azar et al (2023) [19] the temperature pathway is optimized and drops to considerably lower levels beyond 2100. This long-term temperature development has an impact on the social cost of long-lived greenhouse gases since the marginal damage caused by greenhouse gas emissions increases with increasing temperature (see also SM3 where social cost estimates are also presented for two additional temperature pathways).

Furthermore, we find higher SCC estimates than Rennert et al (2022) [34] when using the damage function by Howard & Sterner (2017) [22] but lower when using the damage function by Nordhaus (2018) [27]– in both cases when using similar discount rates. A reason for this is that the main case damage function used by Rennert et al (2022) [34] generates damages that are intermediate between our two cases for similar increases in the global mean surface temperature. There are also other differences in the modeling approach, such as that Rennert et al (2022) [34] use a certainty equivalent approach, while we use a deterministic approach to calculate SCC values. "

[Line 173]: I recommend not to talk about a "link" between GWP and the ratio of social cost – that technically does not seem quite accurate to me. Fundamentally, there are similar concepts at work really.

Author reply: Thank you for the comment. We now write "As illustrated in figure 2 there are similarities between how GWP depends on the time horizon and how the ratio of social costs depends on the discount rate."

[Section 2.4]: I recommend caveating what we can and cannot learn from the analysis of the North Atlantic flights.

Author reply: We have now added the following sentences to over final discussion section: *“The numbers presented here may not be representable for other regions than the North Atlantic region. However, the overall approach adopted in the paper is generalizable to other regions.”*

[Table 1]: I find this Table hard to read and hard to infer meaning from it. It also does not seem quite in line with the text – possibly as the percentile cases are not fundamentally linked to each other? Is it better to just show the ratios and plot a distribution of those?

Author reply: We have decided to delete this given that we now have an additional analysis on rerouting of flights.

[Line 229]: “fFigure” (there are a few instances in the paper where you capitalize / do not capitalize “Figure”. You might want to be consistent).

Author reply: Thank you, fixed.

[Line 277]: “Understanding the implications of heterogeneity is crucial for the design of cost-effective policy instruments” – I would challenge this statement because this world is full of uncertainty in which a policy does not have to be “right” on a flight-by-flight basis. This points to my concerns around uncertainty discussed above.

Author reply: We agree, but we still think the cost-efficiency (or cost-effectiveness) would require good knowledge about the heterogeneity, otherwise the policy would be relatively blunt, but could of course still be better in term of costs than no policy at all. The final discussion section in the paper is now more clearly focused on policy-heterogeneity-uncertainty.

[Lines 323-331] Does this mean that you cannot capture feedbacks between warming of contrails and aviation CO₂? It probably doesn’t matter too much but would be good to understand.

Author reply: We are not sure we fully understand the comment by the reviewer. However, we can at least say that we capture the climate-carbon cycle feedback impacts of contrail cirrus in the estimate of the social cost of energy forcing.

[Lines 326-331]: I am worried about the integration of ERF into DICE. How exactly is this done?

Author reply: As we now first present SCEF and subsequently present SC-contrail in the paper, and have rewritten the method section accordingly, we hope that the explanation on how the output from the different models (M-DICE and CoCIP) are combined is now clearer.

[Lines 346-350]: I very much appreciate the idea of Ramsey discounting, but I think you might want to provide an initial discount rate in the text for calibration.

Author reply: Yes, we have such information in section 1 of the paper: *“A range of cases are analyzed for the social costs: 1) Three different assumptions for the discount rate; a) a low discount rate (about 2%/year) [29], b) a medium discount rate case (about 2.4%/year) [30], c) a high discount rate case (about 4.3%/year) [27].* “ We also now write the following in the method section: *“With an average*

growth rate in per capita consumption of about 1.9%/yr over the 21st century as we get in M-DICE it implies an average discount rate of about 2%/yr in the low discount case, about 2.4%/yr in the medium discount rate case about 4.3%/yr in the higher discount rate case."

Reviewer #2 (Remarks to the Author):

The manuscript "The Social Costs of Aviation: Comparing Contrail Cirrus and CO₂" delves into an essential but often overlooked aspect of aviation's climate impact: contrail cirrus. By comparing the social costs of contrail cirrus with those of CO₂ emissions, the authors provide valuable insights that could inform policy decisions. However, to enhance the robustness and depth of their analysis, several key areas warrant further exploration. The following comments offer specific suggestions for the authors to consider in their revisions.

Author reply: Thank you for reading our paper and for providing feedback

Strengths:

Addresses a knowledge gap: This paper illuminates the often overlooked but significant social costs of contrail cirrus, highlighting an important aspect of aviation's climate impact.

Comparative analysis: By comparing the social costs of CO₂ and contrail cirrus, the paper provides valuable insights to guide policymakers in reducing aviation's overall climate impact.

Emphasizes variability: The study underscores the varying climate effects of contrail cirrus based on different atmospheric conditions.

Examines influencing factors: The research delves into how discount rates, temperature pathways, and contrail ERF estimates affect social cost calculations.

Policy implications: The discussion includes potential cost-effective strategies for mitigating climate impacts, considering the variability of contrail effects.

Author reply: Thank you for these positive words

Points for Improvement:

Methodology transparency: It would be beneficial to offer more detailed explanations of how the DICE model was updated to include contrail cirrus effects.

Author reply: We have now changed the paper in how we presented results, by starting with the presentation of the social cost of an ideal energy forcers (called social cost of Energy Forcing, SCEF) and how that was used for estimating the social cost of contrail cirrus. Also, the method section has been updated accordingly. We believe that this more clearly shows how the output from the two models (M-DICE and CoCIP) are combined.

Alternative damage functions: Consider assessing the impact of using different damage functions, beyond those of Howard & Sterner (2017), to test the robustness of the results.

Author reply: We had such an analysis in the SM of the previous version of the paper, but we have changed the paper so that social cost results are presented for both the damage function by Howard & Sterner (2017) and the original damage function in the DICE version we use, being Nordhaus (2018).

High SCC estimates: Provide a clearer rationale for the higher social cost of carbon (SCC) estimates compared to other studies and discuss the implications for policy-making.

Author reply: Since we now use two different damage functions the overall message on the magnitude of the social costs in the paper is somewhat changed. We now write *“The social cost of CO₂ is strongly dependent on the discount rate and the damage function, and also, but to a lesser extent, the future temperature path [19, 30, 34], see figure 1. Lower discount rates, higher damage costs and higher future temperature pathways increase the SCC. However, when the discount rate is high, the impact of the temperature pathway on the SCC is rather low. When using the damage function in Howard & Sterner (2017) [22] and a low discount rate, the SCC is 860, 1150 and 1600 US\$/ton CO₂, for the 1.5°C, 2°C and 3°C temperature pathways, respectively. For the medium discount rate, the corresponding estimates are 390, 500 and 680 US\$/ton CO₂, while for the high discount rate case, the estimates are 83, 92 and 110 US\$/ton CO₂. When using the damage function from Nordhaus (2018) [27] the social costs are slightly more than a factor of 3 lower for each case, respectively. Hence, spanning between 26 and 500 US\$/ton CO₂ for the cases discussed above. The functional form of the damage functions in [22] and [27] is the same, but the damage for a given level of temperature change above the pre-industrial level is more than a factor three times higher in Howard & Sterner (2017) [22] compared to Nordhaus (2018) [27].*

The estimated SCC values can be compared to estimates in the literature [19, 27, 30, 34, 35]. Especially, our case with a low and medium discount rate and the use of the damage function by Howard & Sterner (2017) [22] gives relatively high SCC values. One reason that we obtain higher SCC values than in Hänsel et al (2020) [30] and Azar et al (2023) [19] is the use of exogenous temperature pathways where the long-term temperature pathways in our analysis stabilize between 1.5°C to 3°C above the pre-industrial level, while in Hänsel et al (2020) [30] and Azar et al (2023) [19] the temperature pathway is optimized and drops to considerably lower levels beyond 2100. This long-term temperature development has an impact on the social cost of long-lived greenhouse gases since the marginal damage caused by greenhouse gas emissions increases with increasing temperature (see also SM3 where social cost estimates are also presented for two additional temperature pathways).

Furthermore, we find higher SCC estimates than Rennert et al (2022) [34] when using the damage function by Howard & Sterner (2017) [22] but lower when using the damage function by Nordhaus (2018) [27]– in both cases when using similar discount rates. A reason for this is that the main case damage function used by Rennert et al (2022) [34] generates damages that are intermediate between our two cases for similar increases in the global mean surface temperature. There are also other differences in the modeling approach, such as that Rennert et al (2022) [34] use a certainty equivalent approach, while we use a deterministic approach to calculate SCC values.

Limitations of GWP comparison: While the connection between GWP and social cost ratios is intriguing, acknowledge the limitations of GWP as a purely physical metric.

Author reply: We now discuss this in more length, and write *“As discussed in the introduction, metrics based on the social costs are further down the cause-effect chain compared to radiative forcing. This increases the relevance of the potential metric, although such an approach will entail larger uncertainties (Fuglestedt et al, 2003, Forster et al, 2021). Still, it is important to keep in mind*

that the uncertainties in the relative metrics are not necessarily much larger for social cost-based approaches since the numerator and denominator covary for the assumptions concerning the discount rate, future climate pathway and the damage function. “

Expand rerouting analysis: Enhance the discussion on rerouting by including more detailed considerations of route changes, fuel efficiency, and technological improvements.

Author reply: We enhance the discussion on these aspects in the introduction of the paper where we now write as follows on mitigation *“Currently analyzed mitigation strategies for reducing contrail cirrus forcing include a reduction of soot emissions through the use of fuels with a lower aromatic content than present fossil jet fuel [11,12, 13] or by adopting engines emitting less soot [14] and through rerouting of flights around contrail forming regions [15, 16, 17].*

To make informed decisions regarding re-routing of flights, fuel choices, engine development and other potential mitigation measures, the climate impacts of different forcers must be made comparable and the uncertainties of the impacts of different mitigation options need to be assessed. Hence, an understanding of how to value the climate impacts of a short-term forcer with an uncertain climate impact such as contrail cirrus vis-a-vis the climate impact of the long-term forcer CO₂ is critical for designing cost-efficient mitigation strategies. Further, for mitigation strategies such as flight planning and rerouting to avoid contrail forming regions it is critical to also understand the heterogeneity of the impacts. For example, it has been shown that contrail forming regions can theoretically be avoided with rerouting at a relatively low fuel penalty [16, 17], although others argue that the scientific uncertainties regarding weather forecasts, contrail formation, evolution and its radiative impacts are too large for implementing such strategies, especially if the strategies comes at a consequence of increasing the CO₂ emissions [8]. “

In the section about rerouting we have chosen to focus on the issues of uncertainty in greater depth (section 2.4 and figure 6). Covering also a deeper analysis of various mitigation options would take the paper too far from the questions we intend to analyze in the paper and include too much scope. We agree that aspects of fuel efficiency and technology improvements and how they interact with the incentive to reroute are interesting, but beyond the scope of the paper.

Policy recommendations: Develop more specific policy recommendations that take into account both the social cost estimates and the variability in contrail cirrus effects.

Author reply: We have rewritten the discussion section the paper which now says: *“The insights provided in this study offer valuable perspectives for the EU’s ongoing efforts to integrate non-CO₂ aviation effects into climate mitigation regulations as well as for other related policy efforts. Starting in 2025, aircraft operators will be required to report estimated non-CO₂ impacts of their operations, and by 2028, the European Commission aims to propose legislation based on these monitored results to mitigate non-CO₂ aviation effects [47]. Understanding the implications of the climate impact of different forcers, their heterogeneity, and uncertainty is essential for designing cost-efficient policy instruments. For example, applying a simple adjustment factor (constant across all flights) to CO₂ emissions to account for non-CO₂ effects would lead to inefficiencies. By acknowledging the variability in warming impacts, policies could be structured to economically incentivize reductions in strongly warming contrails while paying less attention to weakly warming or cooling contrails.*

However, substantial uncertainties and complexities currently hinder estimation of contrail formation and their warming impacts [48, 49]. Effective policy instruments would need both retrospective data on contrail warming effects to verify the climate impact avoided of certain mitigation measures and accurate forecasts of contrail longevity and radiative properties to enable informed mitigation planning. Additionally, any rerouting efforts must comply with air traffic control constraints, adding further complexity to an already challenging strategy. Potential initial policy focus could be on minimizing contrail cirrus effects by targeting high-impact flight slots and routes — such as afternoon and nighttime flights during winter along contrail-prone routes. Policymakers might also consider incentivizing the use of fuels with lower aromatic content under these conditions [13, 50], although there are still uncertainties about how these fuels impact contrail characteristics.

Finally, for contrail cirrus mitigation initiatives, it is critical to determine not only the radiative forcing from the expected contrails, but (1) how to value climate impacts over time and (2) how to weigh certain versus uncertain impacts of potential mitigation strategies. These decisions are inherently value laden. Science alone cannot determine an appropriate discount rate (or cutoff time horizon when estimating GWPs) nor establish the level of certainty on net climate impact required to classify a mitigation measure as effective. Is it sufficient for a policy to reduce impacts on average, or is higher certainty required for each mitigation effort to reduce climate impact? The greater the certainty we require in reducing climate impacts, the more we should prioritize abatement options with quantifiable climate benefits. Similarly, the more we value long-term climate outcomes, the more emphasis we should place on reducing emissions of long-lived greenhouse gases.”

Additional Considerations:

Ongoing mitigation research: Briefly mention current research efforts aimed at reducing contrail formation through operational or technological solutions.

Author comment: We have added this to the introduction as well as in the discussion, see responses to reviewers comments above.

Limitations of the social cost approach: Discuss potential limitations, such as uncertainties inherent in economic models and damage functions.

Author reply: We have now a paragraph in the method section reading: *“The results from IAMs in general, and especially simple IAMs with a modeling time horizon stretching several hundreds of years into the future, should always be interpreted with care. There are many uncertainties, and in this study, we place emphasis on key uncertainties impacting the SCC and SCEF. We explicitly analyze the consequences on the social costs for the choice of discount rate, future climate pathway, the damage function calibration, contrail cirrus forcing uncertainty and heterogeneity. However, there are uncertainties and potential feedbacks within the integrated climate and economic system that we do not account for and the results should be seen as preliminary and subject to revision as knowledge progresses. “*

Overall, the manuscript provides significant insights into the social costs of contrail cirrus and their

implications for aviation policy. By addressing the suggested improvements, the authors can enhance the paper's strength and its contribution to the field.

Author reply: Thank you for your relevant and intriguing comments.

Strengthening the Analysis:

Discount Rate Sensitivity: The paper explores the use of different discount rates, but it would be beneficial to delve deeper into the rationale for including a low rate, such as 2%. Is this rate a realistic representation of societal preferences for future generations? Furthermore, consider the broader implications of using a wider range of discount rates, especially those reflecting a stronger emphasis on future impacts.

Author reply: We have expanded the discussion and related it more strongly to the recent work by Rennert et al which is fundamental for USEPA recent estimates. On this we write *"In this paper, we will use three representative cases for the discount rate parameters, a "low", a "medium" and "high" case. We set $\eta=1$ and $\delta=0.1\%/yr$ in the low case following Stern (2007) [29], in the medium case we set $\eta=1$ and $\delta=0.5\%/yr$ following the "median expert view" in Hänsel et al (2021) [30] and in the high discount case we follow Nordhaus (2018) [27] and set $\eta=1.45$ and $\delta=1.5\%/yr$. The medium assumption is consistent with the median η and δ obtained when surveying the view among economists [45] and philosophers [46]. With an average growth rate in per capita consumption of about $1.9\%/yr$ over the 21st century as we get in M-DICE it implies an average discount rate of about $2\%/yr$ in the low discount case, about $2.4\%/yr$ in the medium discount rate case about $4.3\%/yr$ in the higher discount rate case.*

The calibration of η and δ is largely on par with the assumptions used by Rennert et al (2022) [34]. The assumptions used by Rennert et al (2022) [34] imply, if used together with an average growth rate in per capita consumption of about 1.9% per year over the 21st century, a discount rate of $1.9\%/yr$ in their low case (with $\eta=1.02$ and $\delta=0.01\%/yr$), $3.2\%/yr$ in their main case (with $\eta=1.42$ and $\delta=0.5\%/yr$), and $3.8\%/yr$ in their high case (with $\eta=1.57$ and $\delta=0.8\%/yr$). Given that the work by Rennert et al and USEPA is among the most comprehensive assessment about social costs over the last years we believe our suggest range is suitable. Also, given that our range is not only comparable to the range suggested by Rennert et al (2022) and US EPA (2023) but also based on established research by Stern (2007), Nordhaus (2018), Drupp et al (2018) and Nesje et al (2023) we believe it reflects the discounting values that is in focus in the prevailing academic literature.

Temperature Pathway Dependence: The analysis indicates that the social cost of contrail cirrus varies depending on future temperature pathways. To enhance this aspect, it would be worthwhile to explore whether this dependency becomes more pronounced under higher temperature scenarios. Investigate if more extreme temperature pathways significantly affect the social cost estimates.

Author reply: We have added two additional cases in SM3 (table SM1). A long-term temperature stabilisation at 4 degree C, and a long-term stabilisation at 1 degree C reached by 2150. These results validate the finding in the main part of the paper that the social costs increase with increasing future temperature, and that the ratio of the social cost of a short-lived climate forcer (contrail cirrus) to that of CO₂ drops with increasing future temperatures.

Contrail ERF Uncertainty: While the paper highlights uncertainty by using two different estimates for contrail ERF, this could be further quantified using a probabilistic approach. By doing so, the analysis would provide a clearer picture of how this uncertainty impacts the social cost estimates and what it means for policy decisions.

Author reply: We have now added such an analysis and integrated it into the whole paper. We make a probabilistic assessment of contrail cirrus RF as well as of the efficacy. Further, for the flight specific contrail forcing analysis we also add additional data from COCIP based on 10 ensemble runs of the ERA5 reanalysis which provides input on the metrological conditions used by COCIP. The changes made to the paper related to uncertainty is reflected in the whole paper rather than in any single section.

Distance-Based Social Costs: The introduction of distance-based social costs for both CO₂ and contrail cirrus is a useful metric. However, it may not fully capture the complexities of climate impact that vary with flight paths and atmospheric conditions. It would be helpful to explore these limitations and consider how a more refined metric could offer a better understanding of the climate impacts.

Author reply: Due to complexity of the paper when including analysis of uncertainty in contrail forcing and efficacy in the paper we decided to leave out the distance based analysis from the manuscript. Keeping it just got to complex, and the paper was trying to include too many dimensions. For that reason, we have decided to leave the analysis of distance-based reasoning for future research.

Addressing Potential Issues:

Global Average vs. Specific Flight Analysis: The contrast between global average social costs and the significant heterogeneity observed in specific flights is an important finding. However, the limitations of relying on global averages for policy decisions should be acknowledged. It's crucial to highlight the importance of considering this heterogeneity to develop more effective mitigation strategies.

Author reply: Yes, we more clearly emphasise this now in the final discussions where we write “). *Understanding the implications of the climate impact of different forcers, their heterogeneity, and uncertainty is essential for designing cost-efficient policy instruments. For example, applying a simple adjustment factor (constant across all flights) to CO₂ emissions to account for non-CO₂ effects could lead to inefficiencies. By acknowledging the variability in warming impacts, policies could be structured to economically incentivize reductions in strongly warming contrails while paying less attention to weakly warming or cooling contrails.* “

Rerouting Feasibility: The idea of rerouting flights to avoid contrail formation is intriguing, but it's important to address the practical challenges and limitations. Discuss factors such as air traffic control constraints, potential economic costs, and the feasibility of alternative mitigation strategies to provide a more comprehensive view.

Author reply: Yes, we have now a richer discussion on this both in the introduction and in the discussion section. In the discussion section we now write “*However, substantial uncertainties and*

complexities currently hinder estimation of contrail formation and their warming impacts [48, 49]. Effective policy instruments would need both retrospective data on contrail warming effects to verify the climate impact avoided of certain mitigation measures and accurate forecasts of contrail longevity and radiative properties to enable informed mitigation planning. Additionally, any rerouting efforts must comply with air traffic control constraints, adding further complexity to an already challenging strategy. Potential initial policy focus could be on minimizing contrail cirrus effects by targeting high-impact flight slots and routes — such as afternoon and nighttime flights during winter along contrail-prone routes. Policymakers might also consider incentivizing the use of fuels with lower aromatic content under these conditions [13, 50], although there are still uncertainties about how these fuels impact contrail characteristics”.

Additional Considerations:

Equity Concerns: It’s important to briefly touch upon the equity concerns related to aviation’s climate impact, especially how it disproportionately affects developing countries. This adds a valuable socio-economic perspective to the discussion.

Author reply: We appreciate the reviewer’s suggestion regarding the relevance of equity concerns related to aviation’s climate impact, particularly with respect to its disproportionate effects on developing countries. While we acknowledge that this is an important and valuable socio-economic perspective, we feel that a discussion on equity would extend beyond the primary focus of our paper and potentially dilute its content. Equity considerations in this context are indeed complex and multifaceted, and addressing them adequately would require a more extensive treatment than the scope of this paper allows.

Reviewer #3 (Remarks to the Author):

The paper explores the social costs associated with aviation emissions, focusing specifically on the climate impacts of contrail cirrus and CO₂. It aims to provide a framework for estimating and comparing the social costs of these two contributors to climate change. The authors employ the Dynamic Integrated Climate Economy (DICE) model, considering different discount rates and future climate pathways. They analyse the sensitivity of the social cost of contrail cirrus and CO₂ to these factors and highlight the significant variability in contrail cirrus costs due to specific meteorological conditions.

Author reply: Thank you for reading our paper and providing comments!

Overall, it is my opinion that this paper may not yet be suitable for acceptance. My primary concerns are outlined below:

1a. Lack of a clear definition of the contribution of this paper:

The paper lacks a concise description of its main contribution. While the findings are clear, it is

necessary to provide specific results, especially regarding the significant variation in social costs under different discount rates.

Author reply: Thank you for pointing this out. We believe this is mainly related to the clarity of our messaging which we have now improved. We now write the following in the introduction: *“The aim of this paper is four-fold:*

- 1. Develop a methodology to estimate the social cost of CO₂ (SCC) and contrail cirrus (SC-contrails) in a consistent framework,*
- 2. estimate the SCC and SC-contrails, with a particular focus on the uncertainty and heterogeneity of the latter,*
- 3. compare the ratio of SC-contrail to SCC with the GWP, and*
- 4. examine how the SCC, the SC-contrails, and the role of risk aversion to uncertain climate impacts affect the balance between using more fuel and avoiding contrails.*

The analysis in the paper goes beyond the previous research on the social cost of aviation CO₂ and contrail cirrus in several ways [24, 25, 26]: 1. It contributes with an updated and thorough assessment of the role of the discount rate, the damage function and the future climate pathway for the social cost of carbon and contrail cirrus. 2. It analyzes the heterogeneity in the social cost of contrail cirrus based on half a million flights over the North Atlantic region and 3. It uses the social cost estimates to analyze the implications for rerouting of individual flights when considering the heterogeneity and the uncertainty in contrail cirrus forcing estimate.”

1b. The definition and significance of the discount rate, a crucial aspect of the paper, are not sufficiently explained. Clear definitions and explanations are essential for readers to understand the paper's core concepts.

Author reply: We have now done this by 1) writing the following in the introduction: *“The social cost of a climate forcer is defined as the net present value of future damages caused by an emission pulse of that forcer. This value is influenced by a discount rate, which gradually reduces the weight of impacts occurring further into the future...”* , and 2) adding the following definition on discount rate in the method section of the paper: *“In principle, the use of discounting implies that future climate impacts will be valued gradually less and the social cost of the forcer X emitted in time t₀, denoted SC_X(t₀), will depend on the future climate impact as follows:*

$$SC_X(t_0) = \int_{t_0}^{\infty} \frac{\partial D(t)}{\partial T(t)} \cdot \frac{\partial T(t)}{\partial E_X(t_0)} e^{-rt} dt \quad (2)$$

where D(t) is the climate damages in time t, T(t) the temperature response in time t and E_X(t₀) the emissions of forcer X in time t₀.”

2. Lack of specificity in the importance of comparing the social cost of contrail cirrus and CO₂: The purpose and significance of comparing the social costs of contrail cirrus and CO₂ emissions need to be more explicitly specified. The paper should clearly state why this topic is important and elaborate on the relevance of this comparison within the context of aviation emissions.

Author reply: Thank you for this comment. It is of course vital that this issue is crystal clear to the readers. One of the main policy alternatives is to fly around areas where contrails are formed. This reduces contrails but increases fuel use and carbon emissions. To discuss the trade-offs, we need to be able to compare the two. We have added the following to the introduction: *“Hence, there are fundamental differences in the time dynamics of the climate impacts of CO₂ emissions and contrail cirrus. This temporal difference in climate impacts is critical when evaluating mitigation options and policy instruments for the aviation sector.”* In addition, in the introduction, we further provides a specification of the importance of the social cost based approach: *“The climate impacts of different forcers are typically assessed using emission metrics (Borella et al, 2024). These can either be physics based, e.g., Global Warming Potential (GWP) or be based on economic approaches., e.g., the social cost [18, 19]. The GWP is by far the most widely used emission metric [20]. Its value for a given climate forcer is given by the integrated change in ERF over a chosen time horizon following an emission pulse of that forcer divided by the corresponding estimate for an emission pulse of CO₂ [20].*

A social cost-based approach offers an alternative [18, 19]. The social cost of a climate forcer is defined as the net present value of future damages caused by an emission pulse of that forcer. This value is influenced by a discount rate, which gradually reduces the weight of impacts occurring further into the future. Consequently, the social cost depends on factors such as the discount rate, the damage function, economic growth, and the temperature pathway considered.

In relation to GWP, the social cost-based approach considers additional dimensions that have theoretical appeal for characterizing the marginal climate impact of emissions. 1) It is based on integrated economic damage rather than integrated radiative forcing. Damages are further down the cause-effect chain compared to radiative forcing implying a larger socioeconomic relevance, but also larger uncertainty [21]. 2) It adopts the use of discounting instead of an integration time horizon. Discounting has a stronger welfare-theoretical appeal than the use of an arbitrary time horizon. 3) It takes into account that the marginal impact of one additional emission unit is larger for higher temperature increases. This is consistent with the climate impact assessment literature [22, 23].
“

3. Need for clarification on assumptions about global average conditions and the North Atlantic Region:

The authors mention assumptions related to global average conditions and the comparison with the North Atlantic Region. However, these assumptions require further explanation to provide a comprehensive understanding of the study. Clarifying these assumptions would enhance the transparency and applicability of the research.

Author reply: The reason for focusing in on flights in the North Atlantic region is to apply the methodology on single flights instead of global average conditions. For contrail cirrus energy forcing estimates we rely on CoCIP for both the global analysis and the analysis of individual flights in the North Atlantic region so the assumptions are consistent. With regards to this we now writes the following in the introduction: *“With our modified version of DICE (M-DICE) we estimate the SCC as well as the social cost of energy forcing (SCEF). SCEF is the social cost per GJ energy forcing for an ideal short-lived forcer with an efficacy equal to one and an atmospheric lifetime less than a year.*

These estimates of the social costs are subsequently applied to CO₂ emissions from global aviation and contrail cirrus forcing estimates [6], as well as to the CO₂ emission and energy forcing for individual flights over the North Atlantic region [28].

The uncertainty analysis in the paper is based on the following approaches: 1) An estimate of systematic uncertainties in the energy forcing calculations as well as of the efficacy value of contrail cirrus energy forcing using a probability density function estimated from the literature. This approach is applied for both global contrail cirrus forcing estimates as well as the flight specific forcing estimates. 2) Estimates of how contrail cirrus energy forcing on a flight specific basis depends on uncertainty in inputs on weather and radiation fields. The uncertainties in these fields are based on 10 different ensemble members of the ERA5 reanalysis (HRES) dataset [33]. The spread among the different ensemble members reflects how uncertainties in observations and model parametrization affect the reanalysis output, which in turn impacts contrail cirrus energy forcing estimates [28]. For more details on models, assumptions and uncertainty estimates, see the method section. "

4. Insufficient explanation regarding the innovative contribution and regional comparisons:
The contribution of the paper, as described between lines 93 and 98, appears to be limited to a typical usage in a particular region. To establish a stronger innovative point, it would be beneficial to compare different regions rather than solely focusing on the North Atlantic Region. This would better illustrate the heterogeneity and improve the paper's overall impact.

Author reply: Thank you for pointing this out. We agree that analysing other regions than North Atlantic would be interesting, and we see that as important for a follow up study, but it is beyond the scope of the present study giving the amount of data processing such a study would require. Furthermore, there are substantial uncertainties and heterogeneities within a region such as the North Atlantic Region, and increasing the understanding of the implications of those uncertainties and heterogeneities for cost-efficient policies is important as such. Further, the methodology and the insights provided for the North Atlantic region can in subsequent analysis be applied in other regional contexts or at the global level.

5. Need for more detailed explanations of result numbers:
The paper should provide more comprehensive explanations of the result numbers presented. For instance, between lines 169 and 172, where numbers such as 0.57, 0.33, and 0.094 are listed, it is crucial to clarify the meaning and implications of these values. Providing additional context and interpretation of the result numbers will enhance the reader's understanding.

Author reply: The text has been updated and now reads: "The social cost ratios can be compared to GWP estimates, formally being the ratio of the time-integrated contrail cirrus effective radiative forcing divided by the corresponding forcing of the CO₂ emissions (Forster et al, 2021), while we here instead use the efficacy-adjusted radiative forcing. Hence, GWP is a measure of the relative impact of the time-integrated efficacy adjusted contrail cirrus forcing to the time-integrated forcing per unit emission of CO₂. The impulse response function and radiative efficiency for the atmospheric concentration impact of CO₂ emissions are the same as in IPCC AR6 (Forster et al, 2021). We use three different integration time horizons.

We find that the best estimate GWP for contrail cirrus (measured on a per mass unit CO₂ emission basis) is 0.57 for a 50-year time horizon, 0.33 for a 100-year time horizon and 0.094 for a 500-year time horizon. As illustrated in figure 2 there are similarities between how GWP depends on the time horizon and how the ratio of social costs depends on the discount rate. This holds even though GWP has its roots in physical science and the ratio of social cost has its roots in economics [19, 25, 40, 41, 42, 43]. The fundamental reason for this similarity is that while GWP is based on integration of efficacy-adjusted RF, the social cost is based on an integration of the temperature response to a pulse emission and the fact that there is a close link between the integrated efficacy adjusted RF and the integrated temperature [44]. Furthermore, the time horizon in the GWP measure plays a role that is closely related to the discount rate, or more specifically the inverse of the effective discount rate, where the effective discount rate is defined as the discount rate minus the growth rate [19, 25].

Consequently, using a short time horizon when calculating, for example 20 years, is not consistent with any of the discount rates used here. Further, a prescriptive approach to discounting based on the median view among economists and philosophers [45, 46] suggests that GWP-100 gives too high a relative value for contrail cirrus in a welfare maximizing context.

As discussed in the introduction, metrics based on the social costs are further down the cause-effect chain compared to radiative forcing. This increases the relevance of the potential metric, although such an approach will entail larger uncertainties [20, 21]. Still, it is important to keep in mind that the uncertainties in the relative metrics are not necessarily much larger for social cost-based approaches since the numerator and denominator covary for the assumptions concerning the discount rate, future climate pathway and the damage function. “

6. Lack of explanation regarding the influence of discount rate:

Between lines 239 and 240, the paper briefly mentions the influence of the discount rate without providing sufficient details. It is important to elaborate on why the discount rate is influential and its significance within the context of the study. A more thorough exploration of the discount rate's impact would enhance the paper's credibility and provide a clearer understanding of its implications.

Author reply: We do agree that the discount rate is very important and have worked hard to explain how throughout the whole article. It is definitely one of the main take-aways. The arguments for the sensitivity on the discount rate are the same for the flight-by-flight analysis as on the global level and we believe in section 2.1, 2.2 and 2.3 discuss the importance of the discount rate and why it is so influential at great depth.

7. Request for a more comprehensive explanation of the calculation methods:

The paper should incorporate a detailed explanation of the calculation methods within the main body of the article rather than confining them to the appendix. Presenting the calculation methods in the context of the study would facilitate a better understanding of the research and its findings.

Author reply: We understand the reviewer's point, but the instructions by Nature Communications says that articles should be written as follows: “The main text of an Article should begin with a section headed Introduction of referenced text that expands on the background of the work (some

overlap with the abstract is acceptable), followed by sections headed Results, Discussion (if appropriate) and Methods (if appropriate). (<https://www.nature.com/ncomms/submit/article>). We have our method section after the discussion as suggested by the instructions given by the journal. However, with that said, we have added pieces of text throughout the paper so that the method used and the calculation steps are now more clear in the main part of the paper.

8. Typographical errors:

Several typographical errors are present in the manuscript. These include "CO2and" in line 15, "Ctemperature" in line 148, and a missing "." at the end of a sentence in line 289. Careful proofreading and correction of these errors are necessary to enhance the overall quality of the paper.

Author reply: Thank you for pointing this out. We have now fixed these errors.

9. Concerns regarding the treatment of contrail formation and overestimation:

The authors acknowledge the dependence of the social cost of contrail cirrus on specific meteorological conditions. However, it seems that the paper treats the airline itself as the contrail, which may lead to an overestimate of the influence of contrail cirrus, especially considering the aim of comparing the impacts of contrail and CO₂. Clarifying the approach to contrail formation and its representation in the study would address this concern.

Author reply: We are not sure that we understand the comment by the reviewer. We do not see each contrail as an airline, rather the estimate of contrail cirrus forcing is analysed in a flight-by-flight basis. Each single flight may generate none, one or several contrails along its flight path depending on the route and the metrological conditions prevailing. What we consider in the paper is the total impact on contrail cirrus EF per flight taking into account the route and metrological conditions for each flight.

10. Simplified model and limitations:

While the use of the DICE model is a common practice in integrated assessment modeling, it is important to acknowledge the limitations it introduces. The simplified representation may overlook important factors and feedback loops, potentially resulting in biased or incomplete social cost estimates. Recognizing this limitation is crucial to avoid overreliance on the model's outputs when formulating policies and making decisions.

Author reply: Thank you. Yes, we have added clear description along these lines in the method section reading: *"The results from IAMs in general, and especially simple IAMs with a modeling time horizon stretching several hundreds of years into the future, should always be interpreted with care. There are many uncertainties, and in this study, we place emphasis on key uncertainties impacting the SCC and SCEF. We explicitly analyze the consequences on the social costs for the choice of discount rate, future climate pathway, the damage function calibration, contrail cirrus forcing uncertainty and heterogeneity. However, there are uncertainties and potential feedbacks within the integrated climate and economic system that we do not account for and the results should be seen as preliminary and subject to revision as knowledge progresses."*

REVIEWER COMMENTS

Reviewer #1 (Remarks to the Author):

The paper “The Social Cost of Aviation: Comparing Contrail Cirrus and CO₂” concerns itself with quantifying the social cost of carbon and contrails. I am reviewing the second version of the paper. I remain convinced that the paper is interesting and timely. I recognize that the authors have spent considerable time re-structuring the paper and adding additional analyses, including (but not limited) to address many of my earlier comments. Unfortunately, it is my overall impression that the edits were rushed. The paper now feels incohesive and hard to follow. I strongly recommend that the authors go through the paper with great care to clean up the logic and writing to make the paper accessible. Additional pointers and questions to underpin this overall impression are given in the following:

Authors response: Thank you for the detailed reading of our manuscript and for the elaborate feedback.

1. Uncertainty of contrail impacts

The handling of uncertainty in the paper confuses me. It feels like uncertainty analysis was added around the edges of a modeling framework, which wasn't built to deal with uncertainty. It feels like there is no cohesive approach for addressing uncertainty (but rather piecemeal considerations here and there).

Authors response: Thank you for the comment. It helped us realise the need to present our approach to incorporating contrail forcing uncertainty in greater detail. In response, we have revised the paper to more fully integrate this uncertainty throughout, primarily by introducing our approach more clearly in the introduction and by reframing the discussion of contrail forcing and efficacy uncertainty in the methods section.

That said, we do not believe our approach is incoherent, and we hope the revisions—as well as our responses to the referees' comments below—make this clearer.

For key changes along these lines see lines 180-205, lines 401-560, and lines 827-995 in the manuscript version with tracked changes.

Here are several points which I got hung up in this context (and which led me to the overall impression):

- The authors now model uncertainty in contrail formation, persistence and impacts which is done in a probabilistic framework. The damage function uncertainty and temperature pathway uncertainty are dealt with via a sensitivity analyses. They also consider discount rate (I'd argue that discount rate isn't even subject to uncertainty in a strict sense). The analyses are all in different places and it is quite hard to maintain an overall perspective of the results range given the different methods and approaches. In addition, the overall approach for addressing uncertainty isn't described in a cohesive manner in the paper (this became clear to me when re-reading the last 3 paragraphs of the Introduction).

Authors response: We agree with the reviewer that the discount rate is not an uncertainty in the same sense as radiative forcing uncertainty. In essence the discount rate is an ethical choice. For this reason, we believe it is preferable not to treat the discount rate probabilistically. Instead, we present

it as a set of discrete assumptions, making both the assumptions and their implications for the social costs of different climate forcings transparent to the reader.

Regarding damage functions, while it is possible to treat them probabilistically, the uncertainties in these functions are substantial. We therefore opted to use a few well-known, selected damage functions and assess the implications of each. We believe this is a more transparent and informative approach than assigning probabilistic distributions to such deeply uncertain relationships. Notably, influential work such as the recent update by the US EPA also emphasizes specific damage functions and discount rates rather than treating them probabilistically (see: <https://www.epa.gov/environmental-economics/scghg>).

At the same time, we chose also to keep the temperature pathway explicit, precisely because we wanted to highlight its influence on the resulting social cost estimates. These temperature pathways are generated by the DICE model under constraints on maximum warming. By using a limited number of cases, we can more clearly illustrate how the choice of temperature pathway affects the social cost estimates and the ratio between the social cost of CO₂ and the social cost of contrail cirrus. We believe this is an important insight conveyed in the paper—one that has rarely been explicitly demonstrated in previous literature.

In contrast, for contrail forcing and efficacy, we do apply a probabilistic approach. The reason is that contrail cirrus forcing is both a central focus of the paper and significantly more uncertain than that of CO₂. It is important to emphasize this, the uncertainty as such, and not the “mechanistic” implications of the social costs as done for discount rate, temperature and damage function. We address this uncertainty at the global level by assessing the existing literature and calibrating a gamma distribution accordingly— the details of which are provided in the Methods section.

There are, of course, different valid approaches to integrating climate and economic analysis. We have chosen the approach we believe is most transparent and fit for the purpose of our paper— namely, to explore the relative trade-offs between long-lived CO₂ emissions, with relatively well-characterized radiative forcing, and short-lived and heterogeneous contrail cirrus, with more uncertain forcing under different assumptions on discount rates, temperature pathways and damage functions.

For changes in the text where we make our approach clear see lines 180-205 in the version with tracked changes.

- I appreciate the efforts to model meteorological uncertainties in contrails as well as the efficacy discussion and related uncertainty. However, the framework seems very simplified and might be incomplete. For example, I can't quite see where uncertainty in contrail properties (ice crystals shapes etc.) would be considered.

We acknowledge that our approach is relatively simple, primarily because there are very few studies that perform an uncertainty analysis of effective forcing (EF) caused by individual flights. Platt et al. investigates uncertainty contrail cirrus EF for individual flights, accounting for both weather-related uncertainties and model parameter uncertainties within the CoCiP model (including factors such as ice particle shapes). This is a complex and computationally intensive task beyond explicit treatment in our paper. (for example, Platt et al only analyse 1000 random flights every third day during 2019).

Further, it is important to note that Platt et al. (2024) address only parametric uncertainties within CoCiP and those related to ERA5 input data; they do not consider structural model uncertainties. As a result, the annual forcing uncertainty presented in their study is significantly smaller than that

reported in Lee et al. (2021), which synthesizes results from multiple studies and includes structural uncertainties.

For example, Platt et al. report in their Supplementary Material: "The mean global EF_{pfm} from our study is 20.3 MJ/flight m, with a 90% confidence interval of [15.8, 25.7] MJ/m. This mean is equivalent to a global annual mean contrail net radiative forcing (RF) of 77 [60, 95] mW/m², assuming 61.3 billion flight kilometers per year in 2018, as in Lee et al. (2021)."

In contrast, Lee et al. (2021) estimate a much wider range: 111.4 (33, 189) mW/m², due to the inclusion of structural uncertainties.

Because of these limitations, we have chosen not to perform a parametric uncertainty analysis of individual CoCiP input assumptions. Moreover, developing new CoCiP-based modeling for this paper would be beyond its scope and intention. We believe such detailed modeling efforts are best reserved for studies specifically focused on improving the representation and understanding of contrail forcing uncertainty.

Our aim in this paper is instead to estimate the social cost of contrail forcing using the best available uncertainty estimates from the current literature.

- In the individual flight model, the authors assume the weather model adjustment in addition to the high-level EF adjustment, which seems to be designed to match the form of the adjustment for the global framework. Isn't that inconsistent (in the sense that the different models might already be capturing the meteorological uncertainties?)? Or am I missing something here?

In principle, we agree with the reviewer's comment: weather-related uncertainty is indeed included in our high-level EF adjustment (with high level uncertainty we mean global level uncertainty estimated from Lee et al, which is also used to estimate probabilistic scaling factor used for the EF for individual flights). However, at an aggregate level (i.e., annual averages), the overall impact of ERA5 ensemble variation is minimal. For our focus on North Atlantic flights, this is illustrated in Supplementary Material Table S6 of Teoh et al. (2022), which shows that the annual EF from contrails is largely insensitive to the choice of ensemble member.

Platt et al. (2024) also address this issue in their Supplementary Material, writing: "The confidence interval for EF comes from variation in CoCiP parameters. When averaging the very large number of flights used in this work, uncertainties in the mean EF due to weather cancel out. The 90% confidence interval due to variations between weather ensembles is ± 0.65 MJ/flight m, which is only 4% the width of the uncertainties due to parameters, demonstrating that averaging over an entire year substantially reduces the uncertainty due to the weather."

Thus, while we do account for weather-related uncertainty in our high-level EF adjustment, this does not significantly affect the aggregate results. Therefore, we do not see weather uncertainty as an argument against the validity of our approach.

We believe our global-level uncertainty estimate is well-founded, as it is based on the mode from Teoh et al. (2024) and the 90% confidence interval aligns closely with estimates from Lee et al. (2021). We extend this global uncertainty to the flight-level analysis by assuming that random draws from the high-level uncertainty distribution are the same across all flights. This means:

- The relative uncertainty remains the same at the global, annual level, as for the per flight specific forcing for each specific ensemble member.

- The same applies at the North Atlantic, since weather-related uncertainties (i.e., differences between ERA5 ensemble members) tend to cancel out when aggregated over all flights.
- We account for additional uncertainty, weather uncertainty, at the flight level by incorporating variation across the 10 ERA5 ensemble members.

With this approach, we aim to be as consistent as possible in translating high-level (i.e., global-level) uncertainty to the flight level.

To validate our flight-level uncertainty assumptions, we compare our scaled EF/flight-km values—accounting for both uncertainty through probabilistic scaling and weather ensemble variation—with those reported by Platt et al. (2024). Platt et al. (2024) analyze contrail formation and EF per flight for every third day throughout 2019, using 1,000 randomly selected flights per day.

We extracted all flights in Platt et al (2024) that passes through the same North Atlantic region we have in our datasets and compare flight specific forcing and related uncertainties. Our dataset includes 477,923 flights, compared to 2,309 flights in Platt et al. (2024).

We have added a discussion of this comparison to the Supplementary Material (see page 8 and 9 in the SM). As discussed and shown there, the uncertainty characteristics of EF/flight-km in our analysis align with those found by Platt et al., although—as intended—we have, using our approach a somewhat wider uncertainty interval due to our inclusion of structural uncertainties.

In the SM we write: *“Platt et al. (2024) attempts to estimate contrail forcing uncertainty for individual flights. The study by Platt et al. (2024) considers uncertainties in weather, by using the same 10 ERA5 ensemble members as we do, as well as uncertainties in seven critical parameter assumptions in CoCiP. It analyses uncertainty in contrail formation and energy forcing (EF) by examining 1,000 randomly selected flights every third day during 2019.*

We have extracted EF for all flights in Platt et al. that operate in the same region of the North Atlantic as the flights considered in this paper (Teoh et al., 2022), and compare the uncertainty characteristics to our approach. Hence, we compare the uncertainty in EF per flight-kilometer from the flights included in Platt et al. (2024) to the uncertainty characteristics (in terms of EF per flight-kilometer) of the flights we consider in our paper, when using the 10 different ERA5 ensemble members scaled with probabilistic EF weighting.

In the figure, we show the EF per flight-kilometer for the flights with non-zero EF in our probabilistic sample (blue markers), as well as the EF per flight-kilometer for the flights with non-zero EF in Platt et al. (2024) that pass through the same region. These are plotted against the average contrail cirrus EF caused by the respective flights. The overall pattern of the relationship between average EF per flight-kilometer and its distribution appears relatively similar for our approach and that of Platt et al. However, our approach tends to produce a somewhat wider distribution overall, which is also our intention since we want to also cover the impact of structural uncertainties, not just parametric uncertainties.

Figure SM6. Probabilistic EF realizations plotted against their average values using our approach (blue markers) and the approach by Platt et al. (2024) (red markers).”

- Shouldn't the uncertainties be discussed in a more upfront manner, e.g.: include them in Figure 1, 4 and 5 or in the discussion (e.g., see lines 451 ff in the marked up version)?

Authors response: The uncertainty in contrail radiative forcing (RF) does not affect Figure 1, as this figure presents the social cost per tonne of CO₂ and per gigajoule (GJ) of any climate forcer. Since contrail RF uncertainty is not directly relevant to these normalized metrics, it cannot be reflected in this figure.

However, in response to the reviewer's comment, we have revised Figures 4 and 5 to explicitly incorporate contrail RF uncertainty. As a consequence of these changes, we have removed Figure 6, as its analysis became redundant and did not add significant additional insight beyond what is now captured in the updated Figures 4 and 5 and the associated revised text.

We appreciated the reviewer's comments on uncertainty and they have inspired us to improve these figures. We believe that the figures in the paper now reflect uncertainty in contrails in a better way.

- In Section 2.2, the authors state a total RF number. The methods section then describes how they get to a distribution of that number. That seems inconsistent. How do these relate?

Authors response: We are uncertain which specific numbers the referee is referring to. We have carefully double-checked all figures presented in the manuscript and believe they are correct.

2. Flight-by-flight analysis

The authors add a flight-by-flight contrail analysis. While I generally appreciate that, they then draw conclusions on the benefits of contrail mitigation based on a fixed fuel burn number. I do not think that this is a valid way of looking at these results, given that we know that the fuel burn penalties for contrail avoidance vary among flights.

Authors response: Yes, we agree that fuel burn penalties will vary across individual flights. The fuel burn penalty assumptions were included for illustrative purposes, to provide a first-order understanding of the potential trade-off between contrail avoidance and additional fuel consumption. We now state this explicitly in the text.

Furthermore, we have revised Figure 5 to more clearly emphasize that the 1% and 5% fuel burn assumptions are illustrative examples. In addition, we have removed Figure 6, as the uncertainty is now fully integrated into Figures 4 and 5, and Figure 6 no longer adds any meaningful additional insight.

3. Global metrics

I wonder whether the approach of calculating a cost of contrail RF via a global shadow price can be valid. The DICE model determines shadow prices of CO₂ and short-term forcers. In contrast to CO₂, contrails are not well mixed. There is evidence of regionalized sensitivities. I agree that this is very hard to capture, but wouldn't it be good to discuss this at the very least? It would raise the question how valid it is to apply global cost metrics to the North Atlantic case (which I continue to feel somewhat uncomfortable with).

Authors response: Yes, thanks, the comment is appreciated! We now discuss this in great length in method section where we now write *“We assume that the damage function is applicable to short-lived forcers such as contrail cirrus. The climate response of short-lived forcers does not follow the same pattern as that of long-lived forcers but still cause widespread impacts far beyond the local area where the forcing occurs⁵⁸⁻⁶¹. The global average temperature impact of short-lived forcers is captured by the efficacy value. Further, the climate impacts of short-lived forcers tend to be more localized, with temperature impacts primarily within the hemisphere the emissions occur, and especially strong along the zonal direction^{59,60,62}. Given that the majority of contrail cirrus formed in the northern hemisphere, and the fact that majority of the global population and the global economic activity is in the northern hemisphere it is possible that using the global averaged temperature response of the short-lived forcer, a damage function based on global mean surface temperature and global estimates of GDP might underestimate the social cost of short-lived forcers as contrails cirrus using globally aggregated framework as adopted here.*

Related arguments apply for the contrails formed by flight in the North Atlantic region (more specifically in the region 50°W and 10°W and 40°N and 75°N). Although these contrails maybe formed far from the majority of economic activities the impact of these forcers have likely an impact over large parts of the Northern hemisphere and especially in the zonal direction.

Lund et al⁶³ analyses the distribution of temperature impacts of regional aviation forcers. The North Atlantic region (which is placed in our paper within 50°W and 10°W and 40°N and 75°N) largely falls within “EUR” emission source region⁶³. Contrails sourced in the EUR region show an especially strong temperature impact in the latitude bands 28 °N and 90°N, and is in their paper about twice as large as the global average temperature impact of contrails sourced in EUR region⁶³. “

4. Novelty

I continue to think that the idea of quantifying aviation non-CO2 impacts through monetary metrics and comparing them to CO2 impacts is not new (e.g., Dorbian et al. 2011 and Grobler et al., 2019). I am still missing a clear upfront novelty statement relative to existing work. Some language has been added, but still not acknowledging the aviation-specific work in detail.

Authors response: Thank you for pointing this out—it indeed strengthens the paper to clarify our contributions more explicitly. We now highlight the novelties of our approach upfront in the Introduction, in addition to referencing Dorbian et al., Grobler et al., and Azar & Johansson. These contributions include: the use of a Ramsey discounting framework; the inclusion of climate–carbon cycle feedbacks; the focus on estimating the social cost per unit of efficacy adjusted energy forcing (EF), rather than per kilometer flown or per tonne of CO₂; the application of our analysis both at the global level and on a flight-by-flight basis (the latter, to our knowledge, being novel); and the use of updated estimates for the damage function and for calibrating the Ramsey formula.

We also clarify the role of the temperature pathway for the social cost estimates—an aspect that is often underemphasized in the broader social cost literature and has not been clearly addressed in aviation-specific studies.

For key changes along these lines see line 133-143 in the tracked changes version of the manuscript.

5. Additional questions and comments (if line numbers are given, they reflect line numbers in the marked up version of the paper)

- I may have missed it, but somewhere in the paper it might be good to state that this is all done for quantifying the impacts of a flight today (or in a specific year).

Authors response: We use flight data from 2019 in combination with social cost estimates for the year 2020. In our previous version we used social cost estimates for 2025, but we have changed to 2020 as it makes more sense to use given that we use emission and forcing data for 2019. This change also implies that numbers presented on the paper have been slightly revised.

Now it is clearly stated which year(s) the study focuses on at the end of the Introduction, where we write: *“The aviation CO2 emissions and contrail forcing is based on data representative for 2019, 2020, 2025 with additional uncertainty estimates, while the social cost estimates are for 2020. For more details on models, assumptions and uncertainty estimates, see the method section.”*

- [Line 157] What’s the source for the Paris pathway? It seems to be missing.

Authors response: All temperature pathways are generated internally within DICE during modelling step 1. We hope this is now clearer with the revised illustration of the modelling framework. While the pathways are not externally imposed, they are interpreted as being broadly consistent with a specific narrative (objective), for example reaching the Paris Agreement. Our Paris agreement consistent pathway reaches 1.5°C by 2100 and exhibits a peak temperature well below 2°C. The exact

definition of “well below” can of course be debated, but in our case, the peak is 1.8°C. We have now included a reference to the Paris Agreement. See also figure SM1 and figure SM2 and related text.

- In comparisons, the authors use unspecific language like “higher” or “lower”, without stating how much higher and lower values are. This makes it harder to read and understand the such comparisons [specifically cumbersome in Section 2.1]. Quantify if possible.

Authors response: We understand the reviewer’s point, but we prefer to retain the current presentation. In Section 2.1, we compare our results on social costs with those from other studies. However, both our results and those in the literature are based on a range of differing—and often non-overlapping—assumptions, making a strict, one-to-one comparison difficult. There is no clear way to map the assumptions from one study onto another. For this reason, we believe that a strict comparison would not add significant value.

In addition, a lot of the time when we use the terminology higher, it is in the context of pointing out “direction”, or sensitivity with respect to changes of a variable. For instance, we write: “The higher the discount rate, the larger is the relative role of the short-lived forcings in the total climate impact of aviation...” In this context, we believe it is appropriate to use the term as such.

- The introduction talks about contrail mitigation quite a bit. I propose to scale this back as this is not the focus of the paper.

Authors response: We appreciate the reviewer’s comment. The relatively lengthy discussion on this topic is a result of a request from another reviewer during a previous review round, who asked for a more extensive discussion on context and abatement options in the Introduction. For that reason, we would prefer to retain most of the text. However, we agree that it was somewhat too long and have therefore removed the following sentences: “For example, it has been shown that contrail forming regions can theoretically be avoided with rerouting at a relatively low fuel penalty [16, 17], although others argue that the scientific uncertainties regarding weather forecasts, contrail formation, evolution and its radiative impacts are too large for implementing such strategies, especially if the strategies comes at a consequence of increasing the CO₂ emissions [8].”

- [Lines 186ff]: Note that there are different reference styles (with author in text or just footnote – I’d prefer the former).

Authors response: We have carefully gone through the reference style and tried to make it consistent throughout the manuscript.

- [Lines 192-197]: Isn’t the text repetitive?

Authors response: Thank you, yes, we deleted the second sentence here saying “The functional form of the damage functions in [22] and [27] is the same, but the damage for a given level of temperature change above the pre-industrial level is more than a factor three times higher in Howard & Sterner (2017) [22] compared to Nordhaus (2018) [27].”

- [Lines 256ff]: I recently saw a new publication by Prakash et al. (<https://pubs.rsc.org/en/content/articlehtml/2024/se/d4se00419a>) which also includes total climate costs of aviation. The paper seems to have updates from earlier work by that group (Grobler et al). You may want to look at their numbers and compare them here.

Authors response: Thank you for the link—it's indeed an interesting paper. However, since the social cost estimates used in the referenced study are based on Grobler et al. (2019), which we already cite and compare our results to in SM3, and because the primary focus of the linked paper is on the impacts of technological changes rather than on social costs per se, we have chosen not to include it in our manuscript.

- [Line 680ff] The explanation on how contrails are included in DICE seems short and very hard to understand. In my view, this requires more discussion, as it is one of the key points in the method.

Authors response: We have added an illustration (figure SM1) on how CoCiP and DICE are linked. We hope that the figure clarifies how we couple the models.

- [Lines 752ff & 797ff]: I propose adding more explanation which distribution functions are assumed and why. In its current form, the discussion seems somewhat arbitrary.

Authors response: We have rewritten the text on the RF assumptions and the assumed distributions. We hope that it is clearer now. Further, we also now use a formal approach for fitting the gamma distribution to the data we use, instead of an approach based on ocular inspection of the distribution, see Method section for details (lines 827-995 in the version with tracked changes).

- [Lines 776ff]: Results. Why are they in the methods section?

Authors response: These numbers are not a result from our study, they are results from Teoh et al (2022), which we use as input to our work. We have reformulated the text so that this is hopefully clearer now. To be clear, we do not run CoCiP in in the present study, we only use output from the study by Teoh et al (2022).

Reviewer #2 (Remarks to the Author):

Suggestions for Minor Improvements (Optional)

While the manuscript is now well-developed, the following minor suggestions may further refine the study:

Authors response: Thank you for your positive evaluation.

Consider providing a brief graphical summary or flowchart illustrating the integration of M-DICE and CoCiP models, to enhance accessibility for readers less familiar with modeling frameworks.

Authors response: We have now added an illustration of the model setup. We have presented the figure in SM1, but refer to it in the introduction of the paper.

Expand slightly on the policy implications of high SCC values under low discount rate scenarios, especially for long-term climate policies.

Authors response: The policy implications of higher estimates for the SCC values is that more effort should be put into reducing greenhouse gas emissions. However, in this paper, where the focus is primarily on how to value contrails, we have decided not to go into the policy implications of specific

values for the SCC. That would take the paper into a rather separate direction, but it is good point that we will bring with us for further papers and policy discussions.

Reviewer #3 (Remarks to the Author):

I appreciate the authors' thorough and thoughtful revisions in response to my comments. They have significantly improved the clarity and precision of their manuscript, particularly in defining the study's contribution, elaborating on the role of the discount rate, and refining explanations related to methodology and assumptions.

The authors have effectively addressed my concerns by:

- Clearly articulating the study's main contributions and methodological innovations.
- Providing a well-defined explanation of the discount rate and its impact on the analysis.
- Strengthening the justification for comparing the social cost of contrail cirrus and CO₂ emissions.
- Enhancing discussions regarding regional assumptions and model limitations.
- Offering clearer explanations of key numerical results and their implications.

Additionally, the corrections to typographical errors and the improvements in methodological descriptions make the paper more precise and accessible. Given these revisions, I now find the manuscript suitable for publication in its current form.

Authors response: Thank you for your positive evaluation.

Final comments reviewer 1:

1 – In the Introduction, there are a couple of subsequent paragraphs which enumerate points, almost in bullet form (Lines 88-128). It would be nice if the writing could be revised a little to make this flow a bit better.

Response: We have rewritten this and reduced the bullet form style, see the documents with track changes, page, see page 3.

2 – Introduction, Line 91. The authors say that uncertainty analysis is a “focus” of the paper. I think that’s still too much to say (and the rebuttal letter seems to agree with that assessment). I propose softening the language maybe something along the lines of “estimate SCC and SC-contraails, considering uncertainty along a number of modeling dimensions, as well as heterogeneity”.

Response: We have changed the text accordingly.

3 – Line 191: Revise sentence. Word order seems confused.

Response: The sentence has been rewritten for greater clarity and correct grammar.

4 – Method, Lines 626-637. I would recommend adding one sentence which caveats that ERA5 ensemble analysis would likely not capture any systematic challenges in the underlying data (which is a well-documented concern)

Response: We have added such information, and write: “ERA5 has various systematic challenges for contrail modelling^{65,66}. To reduce some of these limitations, Teoh et al²⁹ apply a methodology to correct for biases in relative humidity.”

I hope we have met all expectations.